# Towards Open Temporal Graph Neural Networks

**Kaituo Feng**
Beijing Institute of Technology
kaituofeng@gmail.com

**Changsheng Li** *
Beijing Institute of Technology
lcs@bit.edu.cn

**Xiaolu Zhang**
Ant Group
yueyin.zxl@antfin.com

**Jun Zhou**
Ant Group
jun.zhoujun@antfin.com

## Abstract

Graph neural networks (GNNs) for temporal graphs have recently attracted increasing attentions, where a common assumption is that the class set for nodes is closed. However, in real-world scenarios, it often faces the open set problem with the dynamically increased class set as the time passes by. This will bring two big challenges to the existing temporal GNN methods: (i) How to dynamically propagate appropriate information in an open temporal graph, where new class nodes are often linked to old class nodes. This case will lead to a sharp contradiction. This is because typical GNNs are prone to make the embeddings of connected nodes become similar, while we expect the embeddings of these two interactive nodes to be distinguishable since they belong to different classes. (ii) How to avoid catastrophic knowledge forgetting over old classes when learning new classes occurred in temporal graphs. In this paper, we propose a general and principled learning approach for open temporal graphs, called OTGNet, with the goal of addressing the above two challenges. We assume the knowledge of a node can be disentangled into class-relevant and class-agnostic one, and thus explore a new message passing mechanism by extending the information bottleneck principle to only propagate class-agnostic knowledge between nodes of different classes, avoiding aggregating conflictive information. Moreover, we devise a strategy to select both important and diverse triad sub-graph structures for effective class-incremental learning. Extensive experiments on three real-world datasets of different domains demonstrate the superiority of our method, compared to the baselines.

## 1 Introduction

Temporal graph (Nguyen et al., 2018) represents a sequence of time-stamped events (e.g. addition or deletion for edges or nodes) (Rossi et al., 2020), which is a popular kind of graph structure in variety of domains such as social networks (Kleinberg, 2007), citations networks (Feng et al., 2022), topic communities (Hamilton et al., 2017), etc. For instance, in topic communities, all posts can be modelled as a graph, where each node represents one post. New posts can be continually added into the community, thus the graph is dynamically evolving. In order to handle this kind of graph structure, many methods have been proposed in the past decade (Wang et al., 2020b; Xu et al., 2020; Rossi et al., 2020; Nguyen et al., 2018; Li et al., 2022). The key to success for these methods is to learn an effective node embedding by capturing temporal patterns based on time-stamped events.

A basic assumption among the above methods is that the class set of nodes is always closed, i.e., the class set is fixed as time passes by. However, in many real-world applications, the class set is open. We still take topic communities as an example, all the topics can be regarded as the class set of nodes for a post-to-post graph. When a new topic is created in the community, it means a new class is involved into the graph. This will bring two challenges to previous approaches: The first problem is the heterophily propagation issue. In an open temporal graph, a node belonging to a new class is often

---

*Corresponding author

linked to a node of old class, as shown in Figure 1. In Figure 1, 'class 2' is a new class, and 'class 1' is an old class. There is a link occured at timestamp $t_4$ connecting two nodes $v_4$ and $v_5$, where $v_4$ and $v_5$ belong to different classes. Such a connection will lead to a sharp contradiction. This is because typical GNNs are prone to learn similar embeddings for $v_4$ and $v_5$ due to their connection (Xie et al., 2020; Zhu et al., 2020), while we expect the embeddings of $v_4$ and $v_5$ to be distinguishable since they belong to different classes. We call this dilemma as heterophily propagation. Someone might argue that we can simply drop those links connecting different class nodes. However, this might break the graph structure and lose information. Thus, how and what to transfer between connected nodes of different classes remains a challenge for open temporal graph.

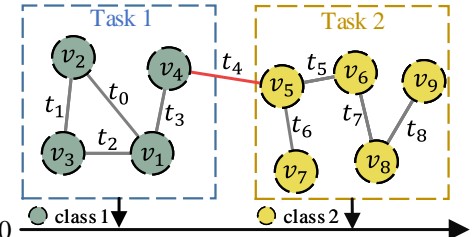

Figure 1: An illustration for an open temporal graph. In the beginning, there is an old class (class 1). As the time passes by, a new class (class 2) occurs. $t_4$ denotes the timestamp the edge is built. The edge occurred at $t_4$ connects $v_4$ and $v_5$ (e.g., the same user comments on both post $v_4$ and post $v_5$ in topic communities).

The second problem is the catastrophic forgetting issue. When learning a new class in an open temporal graph, the knowledge of the old class might be catastrophically forgot, thus degrading the overall performance of the model. In the field of computer vision, many incremental learning methods have been proposed (Wu et al., 2019; Tao et al., 2020), which focus on convolutional neural networks (CNNs) for non-graph data like images. If simply applying these methods to graph-structured data by individually treating each node, the topological structure and the interaction between nodes will be ignored. Recently, Wang et al. (2020a); Zhou & Cao (2021) propose to overcome catastrophic forgetting for graph data. However, They focus on static graph snapshots, and utilize static GNN for each snapshot, thus largely ignoring fine-grained temporal topological information.

In this paper, we put forward the first class-incremental learning approach towards open temporal dynamic graphs, called OTGNet. To mitigate the issue of heterophily propagation, we assume the information of a node can be disentangled into class-relevant and class-agnostic one. Based on this assumption, we design a new message passing mechanism by resorting to information bottleneck (Alemi et al., 2016) to only propagate class-agnostic knowledge between nodes of different classes. In this way, we can well avoid transferring conflictive information. To prevent catastrophic knowledge forgetting over old classes, we propose to select representative sub-graph structures generated from old classes, and incorporate them into the learning process of new classes. Previous works (Zhou et al., 2018; Zignani et al., 2014; Huang et al., 2014) point out triad structure (triangle-shape structure) is a fundamental element of temporal graph and can capture evolution patterns. Motivated by this, we devise a value function to select not only important but also diverse triad structures, and replay them for continual learning. Due to the combinational property, optimizing the value function is NP-hard. Thus, we develop a simple yet effective algorithm to find its approximate solution, and give a theoretical guarantee to the lower bound of the approximation ratio. It is worth noting that our message passing mechanism and triad structure selection can benefit from each other. On the one hand, learning good node embeddings by our message passing mechanism is helpful to select more representative triad structure. On the other hand, selecting representative triads can well preserve the knowledge of old classes and thus is good for propagating information more precisely.

Our contributions can be summarized as : 1) Our approach constitutes the first attempt to investigate open temporal graph neural network; 2) We propose a general framework, OTGNet, which can address the issues of both heterophily propagation and catastrophic forgetting; 3) We perform extensive experiments and analyze the results, proving the effectiveness of our method.

## 2 RELATED WORK

Dynamic GNNs can be generally divided into two groups (Rossi et al., 2020) according to the characteristic of dynamic graph: discrete-time dynamic GNNs (Zhou et al., 2018; Goyal et al., 2018; Wang et al., 2020a) and continuous-time dynamic GNNs (a.k.a. temporal GNNs (Nguyen et al., 2018)) (Rossi et al., 2020; Trivedi et al., 2019). Discrete-time approaches focus on discrete-time dynamic graph that is a collection of static graph snapshots taken at intervals in time, and contains

dynamic information at a very coarse level. Continuous-time approaches study continuous-time dynamic graph that represents a sequence of time-stamped events, and possesses temporal dynamics at finer time granularity. In this paper, we focus on temporal GNNs. We first briefly review related works on temporal GNNs, followed by class-incremental learning.

**Temporal GNNs.** In recent years, many temporal GNNs (Kumar et al., 2019; Wang et al., 2021a; Trivedi et al., 2019) have been proposed. For instance, DyRep (Trivedi et al., 2019) took the advantage of temporal point process to capture fine-grained temporal dynamics. CAW (Wang et al., 2021b) retrieved temporal network motifs to represent the temporal dynamics. TGAT (Xu et al., 2020) proposed a temporal graph attention layer to learn temporal interactions. Moreover, TGN (Rossi et al., 2020) proposed an efficient model that can memorize long term dependencies in the temporal graph. However, all of them concentrate on closed temporal graphs, i.e., the class set is always kept unchanged, neglecting that new classes can be dynamically increased in many real-world applications.

**Class-incremental learning.** Class-incremental learning have been widely studied in the computer vision community (Li & Hoiem, 2017; Wu et al., 2019). For example, EWC (Kirkpatrick et al., 2017) proposed to penalize the update of parameters that are significant to previous tasks. iCaRL (Li & Hoiem, 2017) maintained a memory buffer to store representative samples for memorizing the knowledge of old classes and replaying them when learning new classes. These methods focus on CNNs for non-graph data like images. It is obviously not suitable to directly apply them to graph data. Recently, a few incremental learning works have been proposed for graph data (Wang et al., 2020a; Zhou & Cao, 2021). ContinualGNN (Wang et al., 2020a) proposed a method for closed discrete-time dynamic graph, and trained the model based on static snapshots. ER-GAT (Zhou & Cao, 2021) selected representative nodes for old classes and replay them when learning new tasks. Different from them studying discrete-time dynamic graph, we aim to investigate open temporal graph.

## 3 PROPOSED METHOD

### 3.1 PRELIMINARIES

**Notations.** Let $\mathcal{G}(t) = \{\mathcal{V}(t), \mathcal{E}(t)\}$ denote a temporal graph at time-stamp $t$, where $\mathcal{V}(t)$ is the set of existing nodes at $t$, and $\mathcal{E}(t)$ is the set of existing temporal edges at $t$. Each element $e_{ij}(t_k) \in \mathcal{E}(t)$ represents node $i$ and node $j$ are linked at time-stamp $t_k (t_k \leq t)$. Let $\mathcal{N}_i(t)$ be the neighbor set of node $i$ at $t$. We assume $x_i(t)$ denotes the embedding of node $i$ at $t$, where $x_i(0)$ is the initial feature of node $i$. Let $\mathcal{Y}(t) = \{1, 2, \cdots, m(t)\}$ be the class set of all nodes at $t$, where $m(t)$ denotes the number of existing classes until time $t$.

**Problem formulation.** In our open temporal graph setting, as new nodes are continually added into the graph, new classes can occur, i.e., the number $m(t)$ of classes is increased and thus the class set $\mathcal{Y}(t)$ is open, rather than a closed one like traditional temporal graph. Thus, we formulate our problem as a sequence of class-incremental tasks $\mathcal{T} = \{\mathcal{T}_1, \mathcal{T}_2, \cdots, \mathcal{T}_L, \cdots\}$ in chronological order. Each task $\mathcal{T}_i$ contains one or multiple new classes which are never seen in previous tasks $\{\mathcal{T}_1, \mathcal{T}_2, \cdots, \mathcal{T}_{i-1}\}$. In our new problem setting, the goal is to learn an open temporal graph neural network based on current task $\mathcal{T}_i$, expecting our model to not only perform well on current task but also prevent catastrophic forgetting over previous tasks.

### 3.2 FRAMEWORK

As aforementioned, there are two key challenges in open temporal graph learning: heterophily propagation and catastrophic forgetting. To address the two challenges, we propose a general framework, OTGNet, as illustrated in Figure 2. Our framework mainly includes two modules : A knowledge preservation module is devised to overcome catastrophic

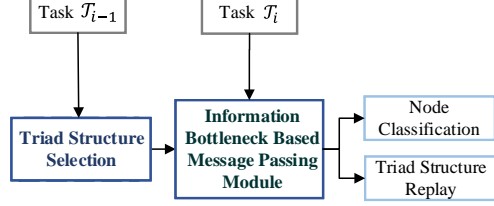

Figure 2: An illustration of overall architecture.

forgetting, which consists of two components: a triad structure selection component is devised to select representative triad structures; a triad structure replay component is designed for replaying the selected triads to avoid catastrophic forgetting. An information bottleneck based message passing module is proposed to propagate class-agnostic knowledge between different class nodes, which can address the heterophily propagation issue. Next, we will elaborate each module of our framework.

## 3.3 KNOWLEDGE PRESERVATION OVER OLD CLASS

When learning new classes based on current task $\mathcal{T}_i$, it is likely for the model to catastrophically forget knowledge over old classes from previous tasks. If we combine all data of old classes with the data of new classes for retraining, the computational complexities will be sharply increased, and be not affordable. Thus, we propose to select representative structures from old classes to preserve knowledge, and incorporate them into the learning process of new classes for replay.

**Triad Structure Selection.** As previous works (Zhou et al., 2018; Zignani et al., 2014; Huang et al., 2014) point out, the triad structure is a fundamental element of temporal graph and its triad closure process could demonstrate the evolution patterns. According to Zhou et al. (2018), the triads have two types of structures: closed triad and open triad, as shown in Figure 3. A closed triad consists of three vertices connected with each other, while an open triad has two of three vertices not connected with each other. The closed triad can be developed from an open triad, and the triad closure process is able to model the evolution patterns (Zhou et al., 2018). Motivated by this point, we propose a new strategy to preserve the knowledge of old classes by selecting representative triad structures from old classes. However, how to measure the 'representativeness' of each triad, and how to select some triads to represent the knowledge of old classes have been not explored so far.

To write conveniently, we omit $t$ for all symbols in this section. Without loss of generality, we denote a closed triad for class $k$ as $g_k^c = (v_s, v_p, v_q)$, where all of three nodes $v_s, v_p, v_q$ belong to class $k$ and $v_s, v_p, v_q$ are pairwise connected, i.e., $e_{sp}(t_i), e_{sq}(t_j), e_{pq}(t_l) \in \mathcal{E}(t_m)$ and $t_i, t_j < t_l, t_m$ is the last time-stamp of the graph. We denote an open triad for class $k$ as $g_k^o = (v_{\tilde{s}}, v_{\tilde{p}}, v_{\tilde{q}})$ with $v_{\tilde{p}}$ and $v_{\tilde{q}}$ not linked to each other in the last observation of the graph, i.e., $e_{\tilde{s}\tilde{p}}(t_i), e_{\tilde{s}\tilde{q}}(t_j) \in \mathcal{E}(t_m)$ and $e_{\tilde{p}\tilde{q}} \notin \mathcal{E}(t_m)$. Assuming $S_k^c = \{g_{k,1}^c, g_{k,2}^c, ..., g_{k,M}^c\}$ and $S_k^o = \{g_{k,1}^o, g_{k,2}^o, ..., g_{k,M}^o\}$ is the selected closed triad set and open

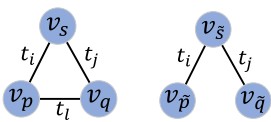

(a) closed triad. (b) open triad.

Figure 3: An illustration for closed triad and open triad.

triad set for class $k$, respectively. $M$ is the memory budget for each class. Next, we introduce how to measure and select closed triads $S_k^c$. It is analogous to open triads $S_k^o$.

In order to measure the 'representativeness' of each triad, one intuitive and reasonable thought is to see how the performance of the model is affected if removing this triad from the graph. However, if we retrain the model once one triad is removed, the time cost is prohibitive. Inspired by the influence function aiming to estimate the parameter changes of the machine learning model when removing a training sample (Koh & Liang, 2017), we extend the influence function to directly estimate the 'representativeness' of each triad structure without retraining, and propose an objective function as:

$$\mathcal{I}_{loss}(g_k^c, \theta) = \left.\frac{\mathrm{d}\,\mathcal{L}(G_k, \theta_{\varepsilon, g_k^c})}{\mathrm{d}\,\varepsilon}\right|_{\varepsilon=0} = \nabla_\theta \mathcal{L}(G_k, \theta)^\top \left.\frac{\mathrm{d}\,\hat{\theta}_{\varepsilon, g_k^c}}{\mathrm{d}\,\varepsilon}\right|_{\varepsilon=0} \quad (1)$$

$$= -\nabla_\theta \mathcal{L}(G_k, \theta)^\top H_\theta^{-1} \nabla_\theta \mathcal{L}(g_k^c, \theta)$$

where $\mathcal{L}$ represents the loss function, e.g., cross-entropy used in this paper. $\theta$ is the parameter of the model, and $G_k$ is the node set of class $k$. $\theta_{\varepsilon, g_k^c}$ is the retrained parameter if we upweight three nodes in $g_k^c$ by $\varepsilon(\varepsilon \to 0)$ during training. $\varepsilon$ is a small weight added on the three nodes of the triad $g_k^c$ in the loss function $\mathcal{L}$. $H_\theta$ is the Hessian matrix. $\nabla_\theta \mathcal{L}(g_k^c, \theta), \nabla_\theta \mathcal{L}(G_k, \theta)$ are the gradients of the loss to $g_k^c$ and $G_k$, respectively. The full derivation of Eq. (1) is in Appendix A.2.

In Eq. (1), $\mathcal{I}_{loss}(g_k^c, \theta)$ estimates the influence of the triad $g_k^c$ on the model performance for class $k$. The more negative $\mathcal{I}_{loss}(g_k^c, \theta)$ is, the more positive influence on model performance $g_k^c$ provides, in other words, the more important $g_k^c$ is. Thus, we define the 'representativeness' of a triad structure as:

$$\mathcal{R}(g_k^c) = -\mathcal{I}_{loss}(g_k^c, \theta) \quad (2)$$

In order to well preserve the knowledge of old classes, we expect all $g_k^c$ in $S_k^c$ are important, and propose the following objective function to find $S_k^c$:

$$S_k^c = \arg \max_{\{g_{k,1}^c, \cdots, g_{k,M}^c\}} \sum_{i=1}^{M} \mathcal{R}(g_{k,i}^c) \quad (3)$$

During optimizing (3), we only take the triad $g_{k,i}^c$ with positive $\mathcal{R}(g_{k,i}^c)$ as the candidate, since $g_{k,i}^c$ with negative $\mathcal{R}(g_{k,i}^c)$ can be thought to be harmful to the model performance. We note that only

optimizing (3) might lead to that the selected $g_{k,i}^c$ have similar functions. Considering this, we hope $S_k^c$ should be not only important but also diverse. To do this, we first define:

$$\mathcal{C}(g_{k,i}^c) = \{g_{k,j}^c | \; ||\bar{x}(g_{k,j}^c) - \bar{x}(g_{k,i}^c)||_2 \leq \delta, g_{k,j}^c \in N_k^c\}, \tag{4}$$

where $\bar{x}(g_{k,j}^c)$ denotes the average embedding of three vertices in $g_{k,j}^c$. $N_k^c$ denotes the set containing all positive closed triads for class $k$, and $\delta$ is a similar radius. $\mathcal{C}(g_{k,i}^c)$ measures the number of $g_{k,j}^c$, where the distance of $\bar{x}(g_{k,j}^c)$ and $\bar{x}(g_{k,i}^c)$ is less or equal to $\delta$. To make the selected triads $S_k^c$ diverse, we also anticipate that $\{\mathcal{C}(g_{k,1}^c), \cdots, \mathcal{C}(g_{k,M}^c)\}$ can cover different triads as many as possible by:

$$S_k^c = \arg \max_{\{g_{k,1}^c, \cdots, g_{k,M}^c\}} \frac{|\bigcup_{i=1}^M \mathcal{C}(g_{k,i}^c)|}{|N_k^c|} \tag{5}$$

Finally, we combine (5) with (3), and present the final objective function for triad selection as:

$$S_k^c = \arg \max_{\{g_{k,1}^c, \cdots, g_{k,M}^c\}} F(S_k^c) = \arg \max_{\{g_{k,1}^c, \cdots, g_{k,M}^c\}} \left( \sum_{i=1}^M \mathcal{R}(g_{k,i}^c) + \gamma \frac{|\bigcup_{i=1}^M \mathcal{C}(g_{k,i}^c)|}{|N_k^c|} \right) \tag{6}$$

where $\gamma$ is a hyper-parameter. By (6), we can select not only important but also diverse triads to preserve the knowledge of old classes.

Due to the combinatorial property, solving (6) is NP-hard. Fortunately, we show that $F(S_k^c)$ satisfies the condition of monotone and submodular. The proof can be found in Appendix A.3. Based on this property, (6) could be solved by a greedy algorithm (Pokutta et al., 2020) with an approximation ratio guarantee, by the following Theorem 1 (Krause & Golovin, 2014).

**Theorem 1.** Assuming our value function $F : 2^N \to \mathbb{R}_+$ is monotone and submodular. If $S_k^{c*}$ is an optimal triad set and $S_k^c$ is a triad set selected by the greedy algorithm (Pokutta et al., 2020), then $F(S_k^c) \geq (1 - \frac{1}{e})F(S_k^{c*})$ holds.

By Theorem 1, we can greedily select closed triads as in Algorithm 1. As aforementioned, the open triad set $S_k^o$ can be chosen by the same method. The proof of Theorem 1 can be found in Krause & Golovin (2014).

---

**Algorithm 1** Representative triad selection

**Input:** all triads $N_k^c$ for class $k$, budget $M$;
**Output:** representative triad set $S_k^c$;
1: Initialize $S_k^c = \emptyset$;
2: **while** $|S_k^c| < M$ **do**
3:     $u = argmax_{u \in N_k^c \setminus S_k^c} F(S_k^c \cup \{u\})$;
4:     $S_k^c = S_k^c \cup u$;
5: **end while**
6: return $S_k^c$

---

**An Acceleration Solution.** We first provide the time complexity analysis of triad selection. When counting triads for class $k$, we first enumerate the edge that connects two nodes $v_s$ and $v_d$ of class $c$. Then, for each neighbor node of $v_s$ that belongs to class $k$, we check whether this neighbor node links to $v_d$. If this is the case and the condition of temporal order is satisfied, these three nodes form a closed triad, otherwise these three nodes form an open triad. Thus, a rough upper bound of the number of closed triads in class $k$ is $O(d_k|\mathcal{E}_k|)$, where $|\mathcal{E}_k|$ is the number of edges between two nodes of class $k$, and $d_k$ is the max degree of nodes of class $k$. When selecting closed triads, finding a closed triad that maximizes the value function takes $O(|N_k^c|^2)$, where $|N_k^c|$ is the number of positive closed triads in class $k$. Thus, it is of order $O(M|N_k^c|^2)$ for selecting the closed triad set $S_k^c$, where $M$ is the memory budget for each class. The time complexity for selecting the open triads is the same.

To accelerate the selection process, a natural idea is to reduce $N_k^c$ by only selecting closed triads from $g_k^c$ with large values of $\mathcal{R}(g_k^c)$. Specifically, we sort the closed triad $g_k^c$ based on $\mathcal{R}(g_k^c)$, and use the top-$K$ ones as the candidate set $N_k^c$ for selection. The way for selecting open triads is the same.

**Triad Structure Replay.** After obtaining representative closed and open triad sets, $S_k^c$ and $S_k^o$, we will replay these triads from old classes when learning new classes, so as to overcome catastrophic forgetting. First, we hope the model is able to correctly predict the labels of nodes from the selected triad set, and thus use the cross entropy loss $\mathcal{L}_{ce}$ for each node in the selected triad set.

Moreover, as mentioned above, the triad closure process can capture the evolution pattern of a dynamic graph. Thus, we use the link prediction loss $\mathcal{L}_{link}$ to correctly predict the probability whether two nodes are connected based on the closed and open triads, to further preserve knowledge:

$$\mathcal{L}_{link} = -\frac{1}{N_c} \sum_{i=1}^{N_c} \log(\sigma(x_p^i(t)^\top x_q^i(t))) - \frac{1}{N_o} \sum_{i=1}^{N_o} \log(1 - \sigma(\tilde{x}_p^i(t)^\top \tilde{x}_q^i(t))), \tag{7}$$

where $N_c, N_o$ are the number of closed, open triads respectively, where $N_c = N_o = N_t * M$. $N_t$ is the number of old classes. $\sigma$ is the sigmoid function. $x_p^i(t), x_q^i(t)$ are the embeddings of $v_p, v_q$ of the $i^{th}$ closed triad. $\tilde{x}_p^i(t), \tilde{x}_q^i(t)$ are the embeddings of $v_{\tilde{p}}, v_{\tilde{q}}$ of the $i^{th}$ open triad. Here the closed triads and open triads serve as postive samples and negative samples, respectively.

### 3.4 MESSAGE PASSING VIA INFORMATION BOTTLENECK

When new class occurs, it is possible that one edge connects one node of the new class and one node of an old class, as shown in Figure 1. To avoid aggregating conflictive knowledge between nodes of different classes, one intuitive thought is to extract class-agnostic knowledge from each node, and transfer the class-agnostic knowledge between nodes of different classes To do this, we extend the information bottleneck principle to obtain a class-agnostic representation for each node.

**Class-agnostic Representation.** Traditional information bottleneck aims to learn a representation that preserves the maximum information about the class while has minimal mutual information with the input (Tishby et al., 2000). Differently, we attempt to extract class-agnostic representations from an opposite view, i.e., we expect the learned representation has minimum information about the class, but preserve the maximum information about the input. Thus, we propose an objective function as:

$$J_{IB} = \min_{Z(t)} I(Z(t), Y) - \beta I(Z(t), X(t)), \tag{8}$$

where $\beta$ is the Lagrange multiplier. $I(\cdot, \cdot)$ denotes the mutual information. $X(t), Z(t)$ are the random variables of the node embeddings and class-agnostic representations at time-stamp $t$. $Y$ is the random variable of node label. In this paper, we adopt a two-layer MLP for mapping $X(t)$ to $Z(t)$.

However, directly optimizing (8) is intractable. Thus, we utilize CLUB (Cheng et al., 2020) to estimate the upper bound of $I(Z(t), Y)$ and utilize MINE (Belghazi et al., 2018) to estimate the lower bound of $I(Z(t), X(t))$. Thus, the upper bound of our objective could be written as:

$$J_{IB} \leq \mathcal{L}_{IB} = \mathbb{E}_{p(Z(t),Y)}[\log q_\mu(y|z(t))] - \mathbb{E}_{p(Z(t))}\mathbb{E}_{p(Y)}[\log q_\mu(y|z(t))]$$
$$- \beta(\sup_\psi \mathbb{E}_{p(X(t),Z(t))}[T_\psi(x(t),z(t))] - \log(\mathbb{E}_{p(X(t))p(Z(t))}[e^{T_\psi(x(t),z(t))}])). \tag{9}$$

where $z(t), x(t), y$ are the instances of $Z(t), X(t), Y$ respectively. $T_\psi : \mathcal{X} \times \mathcal{Z} \to \mathbb{R}$ is a neural network parametrized by $\psi$. Since $p(y|z(t))$ is unknown, we introduce a variational approximation $q_\mu(y|z(t))$ to approximate $p(y|z(t))$ with parameter $\mu$. By minimizing this upper bound $\mathcal{L}_{IB}$, we can obtain an approximation solution to Eq. (8). The derivation of formula (9) is in Appendix A.1.

It is worth noting that $z_i(t)$ is an intermediate variable as the class-agnostic representation of node $i$. We only use $z_i(t)$ to propagate information to other nodes having different classes from node $i$. If one node $j$ has the same class with node $i$, we still use $x_i(t)$ for information aggregation of node $j$, so as to avoid losing information. In this way, the heterophily propagation issue can be well addressed.

**Message Propagation.** In order to aggregate temporal information and topological information in temporal graph, many information propagation mechanism have been proposed (Rossi et al., 2020; Xu et al., 2020). Here, we extend a typical mechanism proposed in TGAT (Xu et al., 2020), and present the following way to learn the temporal attention coefficient as:

$$a_{ij}(t) = \frac{\exp(([x_i(t)||\Phi(t-t_i)]W_q)^\top([h_j(t)||\Phi(t-t_j)]W_p))}{\sum_{l \in \mathcal{N}_i(t)} \exp(([x_i(t)||\Phi(t-t_i)]W_q)^\top([h_l(t)||\Phi(t-t_l)]W_p))} \tag{10}$$

where $\Phi$ is a time encoding function proposed in TGAT. $||$ represents the concatenation operator. $W_p$ and $W_q$ are two learnt parameter matrices. $t_i$ is the time of the last interaction of node $i$. $t_j$ is the time of the last interaction between node $i$ and node $j$. $t_l$ is the time of the last interaction between node $i$ and node $l$. Note that we adopt different $h_l(t)$ from that in the original TGAT, defined as:

$$h_l(t) = \begin{cases} x_l(t), & y_i = y_l \\ z_l(t), & y_i \neq y_l \end{cases}, \tag{11}$$

where $h_l(t)$ is the message produced by neighbor node $l \in \mathcal{N}_i(t)$. If node $l$ and $i$ have different classes, we leverage its class-agnostic representation $z_l(t)$ for information aggregation of node $i$, otherwise we directly use its embedding $x_l(t)$ for aggregating. Note that our method supports multiple layers of network. We do not use the symbol of the layer only for writing conveniently.

Finally, we update the embedding of node $i$ by aggregating the information from its neighbors:

$$x_i(t) = \sum_{j \in \mathcal{N}_i(t)} a_{ij}(t) W_h h_j(t),\qquad(12)$$

where $W_h$ is a learnt parameter matrix for message aggregation.

### 3.5 Optimization

During training, we first optimize the information bottleneck loss $\mathcal{L}_{IB}$. Then, we minimize $\mathcal{L} = \mathcal{L}_{ce} + \rho \mathcal{L}_{link}$, where $\rho$ is the hyper-parameter and $\mathcal{L}_{ce}$ is the node classification loss over both nodes of new classes and that of the selected triads. We alternatively optimize them until convergence. The detailed training procedure and pseudo-code could be found in Appendix A.5.

In testing, we extract an corresponding embedding of a test node by assuming its label to be the one that appears the most times among its neighbor nodes in the training set, due to referring to extracting class-agnostic representations. After that, we predict the label of test nodes based on the extracted embeddings.

## 4 Experiments

### 4.1 Experiment Setup

**Datasets.** We construct three real-world datasets to evaluate our method: Reddit (Hamilton et al., 2017), Yelp (Sankar et al., 2020), Taobao (Du et al., 2019). In Reddit, we construct a post-to-post graph. Specifically, we treat posts as nodes and treat the subreddit (topic community) a post belongs to as the node label.

Table 1: Dataset Statistics

|  | Reddit | Yelp | Taobao |
|---|---|---|---|
| # Nodes | 10845 | 15617 | 114232 |
| # Edges | 216397 | 56985 | 455662 |
| # Total classes | 18 | 15 | 90 |
| # Timespan | 6 months | 5 years | 6 days |
| # Tasks | 6 | 5 | 3 |
| # Classes per task | 3 | 3 | 30 |
| # Timespan per task | 1 month | 1 year | 2 days |

When a user comments two posts with the time interval less or equal to a week, a temporal edge between the two nodes will be built. We regard the data in each month as a task, where July to December in 2009 are used. In each month, we sample 3 large communities that do not appear in previous months as the new classes. For Yelp dataset, we construct a business-to-business temporal graph from 2015 to 2019 in the same way as Reddit. For Taobao dataset, we construct an item-to-item graph in the same way as Reddit in a 6-days promotion season of Taobao. Table 1 summarizes the statistics of these datasets. More information about datasets could be found in Appendix A.4.

**Experiment Settings.** For each task, we use $80\%$ nodes for training, $10\%$ nodes for validation, $10\%$ nodes for testing. We use two widely-used metrics in class-incremental learning to evaluate our method (Chaudhry et al., 2018; Bang et al., 2021): AP and AF. Average Performance (AP) measures the average performance of a model on all previous tasks. Here we use accuracy to measure model performance. Average Forgetting (AF) measures the decreasing extent of model performance on previous tasks compared to the best ones. More implementation details is in Appendix A.6.

**Baselines.** First, we compare with three incremental learning methods based on static GNNs: ER-GAT (Zhou & Cao, 2021), TWC-GAT (Liu et al., 2021) and ContinualGNN (Wang et al., 2020a). For ER-GAT and TWC-GAT, we use the final state of temporal graph as input in each task. Since ContinualGNN is based on snapshots, we split each task into 10 snapshots. In addition, we combine three representative temporal GNN (TGAT (Xu et al., 2020), TGN (Rossi et al., 2020), TREND (Wen & Fang, 2022)) and three widely-used class-incremental learning methods in computer vision (EWC (Kirkpatrick et al., 2017), iCaRL (Rebuffi et al., 2017), BiC (Wu et al., 2019)) as baselines. For our method, we set $M$ as 10 on all the datasets.

### 4.2 Results and Analysis

**Overall Comparison.** As shown in Table 2, our method outperform other methods by a large margin. The reasons are as follows. For the first three methods, they are all based on static GNN that can not capture the fine-grained dynamics in temporal graph. TGN, TGAT and TREND are three dynamic GNNs with fixed class set. When applying three typical class-incremental learning methods to TGN,

Table 2: Comparisons (%) of our method with baselines. The bold represents the best in each column.

| Method | Reddit | | Yelp | | TaoBao | |
|---|---|---|---|---|---|---|
| | AP($\uparrow$) | AF($\downarrow$) | AP($\uparrow$) | AF($\downarrow$) | AP($\uparrow$) | AF($\downarrow$) |
| ContinualGNN | $52.17 \pm 2.46$ | $25.59 \pm 5.39$ | $49.73 \pm 0.27$ | $28.76 \pm 1.52$ | $58.39 \pm 0.24$ | $47.03 \pm 0.50$ |
| ER-GAT | $52.03 \pm 2.59$ | $22.67 \pm 3.30$ | $62.05 \pm 0.70$ | $18.91 \pm 1.09$ | $70.09 \pm 0.88$ | $23.24 \pm 0.36$ |
| TWC-GAT | $52.88 \pm 0.53$ | $19.60 \pm 3.64$ | $60.90 \pm 3.74$ | $16.92 \pm 0.63$ | $59.91 \pm 1.71$ | $42.78 \pm 1.39$ |
| TGAT | $48.47 \pm 1.81$ | $31.03 \pm 4.48$ | $64.89 \pm 1.27$ | $27.31 \pm 3.99$ | $60.62 \pm 0.23$ | $43.35 \pm 0.77$ |
| TGAT+EWC | $50.16 \pm 2.45$ | $28.27 \pm 4.00$ | $66.58 \pm 3.11$ | $25.48 \pm 1.75$ | $64.03 \pm 0.62$ | $38.26 \pm 1.20$ |
| TGAT+iCaRL | $54.50 \pm 2.04$ | $27.66 \pm 1.11$ | $71.71 \pm 2.48$ | $17.56 \pm 2.46$ | $73.74 \pm 1.40$ | $23.90 \pm 2.04$ |
| TGAT+BiC | $54.61 \pm 0.89$ | $25.42 \pm 2.72$ | $74.73 \pm 3.54$ | $16.42 \pm 4.41$ | $74.05 \pm 0.48$ | $23.27 \pm 0.65$ |
| TGN | $47.49 \pm 0.48$ | $32.06 \pm 1.91$ | $56.24 \pm 1.65$ | $41.27 \pm 2.30$ | $65.89 \pm 1.20$ | $36.15 \pm 1.55$ |
| TGN+EWC | $49.45 \pm 1.45$ | $31.74 \pm 1.11$ | $60.83 \pm 3.55$ | $35.73 \pm 3.48$ | $68.89 \pm 2.09$ | $32.08 \pm 3.88$ |
| TGN+iCaRL | $50.86 \pm 4.83$ | $31.01 \pm 2.78$ | $73.34 \pm 1.99$ | $15.43 \pm 0.93$ | $77.42 \pm 0.80$ | $19.57 \pm 1.29$ |
| TGN+BiC | $53.16 \pm 1.53$ | $26.83 \pm 0.95$ | $73.98 \pm 2.07$ | $16.79 \pm 2.90$ | $77.40 \pm 0.80$ | $18.63 \pm 1.69$ |
| TREND | $49.61 \pm 2.92$ | $28.68 \pm 4.20$ | $57.28 \pm 2.83$ | $37.48 \pm 3.26$ | $61.02 \pm 0.16$ | $42.44 \pm 0.14$ |
| TREND+EWC | $53.12 \pm 3.30$ | $25.70 \pm 3.08$ | $65.45 \pm 4.79$ | $26.80 \pm 4.98$ | $62.72 \pm 1.18$ | $40.00 \pm 2.09$ |
| TREND+iCaRL | $52.53 \pm 3.67$ | $30.63 \pm 0.18$ | $69.93 \pm 5.55$ | $15.81 \pm 7.48$ | $74.49 \pm 0.05$ | $23.27 \pm 0.25$ |
| TREND+BiC | $54.22 \pm 0.56$ | $22.42 \pm 3.15$ | $71.15 \pm 2.42$ | $12.78 \pm 5.12$ | $75.13 \pm 1.06$ | $21.70 \pm 0.63$ |
| OTGNet (Ours) | $\mathbf{73.88} \pm 4.55$ | $\mathbf{19.25} \pm 5.10$ | $\mathbf{83.78} \pm 1.06$ | $\mathbf{4.98} \pm 0.46$ | $\mathbf{79.92} \pm 0.12$ | $\mathbf{12.82} \pm 0.61$ |

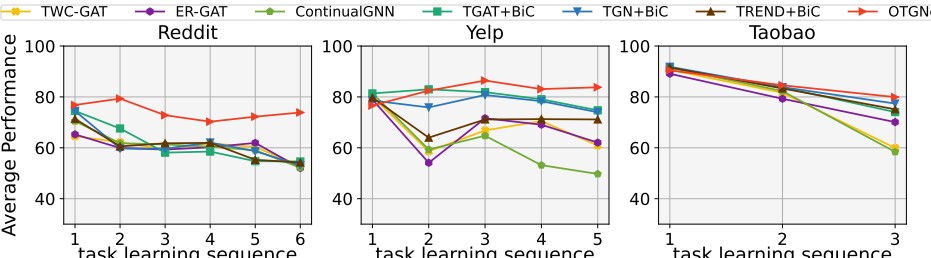

Figure 4: The changes of average performance (AP) (%) on three datasets with the increased tasks.

TGAT and TREND, the phenomenon of catastrophic forgetting is alleviative. However, they still suffer from the issue of heterophily propagation.

**Performance Analysis of Different Task Numbers.** To provide further analysis of our method, we plot the performance changes of different methods along with the increased tasks. As shown in Figure 4, our method generally achieves better performance than baselines as the task number increases. Since BiC based methods achieve better performance based on Table 2, we do not report the results of the other two incremental learning based methods. In addition, the curves of OTGNet are smoother that those of other methods, which indicates our method can well address the issue of catastrophic forgetting. Because of space limitation, we provide the curves of AF in Appendix A.7.

**Ablation Study of our proposed propagation mechanism.** We further study the effectiveness of our information bottleneck based message propagation mechanism. OTGNet-w.o.-IB represents our method directly transferring the embeddings of neighbor nodes instead of class-agnostic representations. OTGNet-w.o.-prop denotes our method directly dropping the links between nodes of different classes. We take GBK-GNN (Du et al., 2022) as another baseline, where GBK-GNN originally handles the heterophily for static graph. For a fair comparison, we modify GBK-GNN to an open temporal graph: Specifically, we create two temporal message propagation modules with separated parameters as the two kernel feature transformation matrices in GBK-GNN. We denote

Table 3: Ablation study of our proposed information bottleneck based propagation mechanism.

| Setting | Reddit | | Yelp | | TaoBao | |
|---|---|---|---|---|---|---|
| | AP($\uparrow$) | AF($\downarrow$) | AP($\uparrow$) | AF($\downarrow$) | AP($\uparrow$) | AF($\downarrow$) |
| OTGNet-w.o.-IB | $54.10 \pm 2.01$ | $34.00 \pm 1.63$ | $76.93 \pm 5.14$ | $14.96 \pm 5.61$ | $79.00 \pm 0.37$ | $13.41 \pm 0.57$ |
| OTGNet-w.o.-prop | $54.67 \pm 2.05$ | $28.73 \pm 2.63$ | $75.67 \pm 1.69$ | $12.87 \pm 1.19$ | $79.07 \pm 0.02$ | $14.48 \pm 0.34$ |
| OTGNet-GBK | $58.79 \pm 1.08$ | $25.22 \pm 2.22$ | $77.03 \pm 2.99$ | $9.79 \pm 1.15$ | $77.73 \pm 0.27$ | $15.49 \pm 0.34$ |
| OTGNet | $\mathbf{73.88} \pm 4.55$ | $\mathbf{19.25} \pm 5.10$ | $\mathbf{83.78} \pm 1.06$ | $\mathbf{4.98} \pm 0.46$ | $\mathbf{79.92} \pm 0.12$ | $\mathbf{12.82} \pm 0.61$ |

Table 4: Results of triad selection strategy on the three datasets.

| Setting | Reddit | | Yelp | | TaoBao | |
|---|---|---|---|---|---|---|
| | AP($\uparrow$) | AF($\downarrow$) | AP($\uparrow$) | AF($\downarrow$) | AP($\uparrow$) | AF($\downarrow$) |
| OTGNet-w.o.-triad | $60.81 \pm 4.46$ | $34.94 \pm 4.73$ | $69.28 \pm 1.73$ | $23.79 \pm 1.75$ | $67.05 \pm 0.44$ | $31.44 \pm 0.41$ |
| OTGNet-random | $69.66 \pm 3.81$ | $23.24 \pm 3.83$ | $78.76 \pm 2.62$ | $9.19 \pm 1.65$ | $79.09 \pm 0.36$ | $13.89 \pm 0.45$ |
| OTGNet-w.o.-diversity | $71.06 \pm 5.73$ | $22.96 \pm 6.91$ | $80.76 \pm 2.60$ | $9.91 \pm 3.83$ | $78.84 \pm 0.46$ | $13.87 \pm 1.18$ |
| OTGNet | $\mathbf{73.88} \pm 4.55$ | $\mathbf{19.25} \pm 5.10$ | $\mathbf{83.78} \pm 1.06$ | $\mathbf{4.98} \pm 0.46$ | $\mathbf{79.92} \pm 0.12$ | $\mathbf{12.82} \pm 0.61$ |

Table 5: Results of evolution pattern preservation on the three datasets.

| Setting | Reddit | | Yelp | | TaoBao | |
|---|---|---|---|---|---|---|
| | AP($\uparrow$) | AF($\downarrow$) | AP($\uparrow$) | AF($\downarrow$) | AP($\uparrow$) | AF($\downarrow$) |
| OTGNet-w.o.-pattern | $70.23 \pm 5.56$ | $23.10 \pm 7.44$ | $81.44 \pm 1.38$ | $6.97 \pm 3.10$ | $79.01 \pm 0.19$ | $14.05 \pm 0.46$ |
| OTGNet | $\mathbf{73.88} \pm 4.55$ | $\mathbf{19.25} \pm 5.10$ | $\mathbf{83.78} \pm 1.06$ | $\mathbf{4.98} \pm 0.46$ | $\mathbf{79.92} \pm 0.12$ | $\mathbf{12.82} \pm 0.61$ |

Table 6: Results of of our acceleration solution with different $K$.

| | Reddit | | | Yelp | | | Taobao | | |
|---|---|---|---|---|---|---|---|---|---|
| | AP($\uparrow$) | AF($\downarrow$) | Time (h) | AP($\uparrow$) | AF($\downarrow$) | Time (h) | AP($\uparrow$) | AF($\downarrow$) | Time (h) |
| $K$=1000 | 73.88 | 19.25 | 1.23 | 83.78 | 4.98 | 0.25 | 79.92 | 12.82 | 1.61 |
| $K$=500 | 71.26 | 22.45 | 0.45 | 83.48 | 6.32 | 0.07 | 79.19 | 13.94 | 0.53 |
| $K$=200 | 66.83 | 26.88 | 0.07 | 81.86 | 6.87 | 0.02 | 79.14 | 13.73 | 0.10 |
| $K$=100 | 66.22 | 28.86 | 0.04 | 78.83 | 10.54 | 0.01 | 78.81 | 14.58 | 0.04 |

this baseline as OTGNet-GBK. As shown in Table 3, OTGNet outperforms OTGNet-w.o.-IB and OTGNet-GBK on the three datasets. This illustrates that it is effective to extract class-agnostic information for addressing the heterophily propagation issue. OTGNet-w.o.-prop generally performs better than OTGNet-w.o.-IB. This tells us that it is inappropriate to directly transfer information between two nodes of different classes. OTGNet-w.o.-prop is inferior to OTGNet, which means that the information is lost if directly dropping the links between nodes of different nodes. An interesting phenomenon is AF score decreases much without using information bottleneck. This indicates that learning better node embeddings by our message passing module is helpful to triad selection.

**Triad Selection Strategy Analysis.** First, we design three variants to study the impact of our triad selection strategy. OTGNet-w.o.-triad means our method does not use any triad (i.e. $M = 0$). OTGNet-random represents our method selecting triads randomly. OTGNet-w.o.-diversity means our method selecting triads without considering the diversity. As shown in Table 4, The performance of our method decreases much when without using triads, which shows the effectiveness of using triads to prevent catastrophic forgetting. OTGNet achieves better performance than OTGNet-random and OTGNet-w.o.-diversity, indicating the proposed triads selection strategy is effective.

**Evolution Pattern Preservation Analysis.** We study the effectiveness of evolution pattern preservation. OTGNet-w.o.-pattern represents our method without evolution pattern preservation (i.e. $\rho = 0$). As shown in Table 5, OTGNet has superior performance over OTGNet-w.o.-pattern, which illustrates the evolution pattern preservation is beneficial to memorize the knowledge of old classes.

**Acceleration Performance of Triad Selection.** As stated aforementioned, to speed up the triad selection, we can sort triads $g_k^c$ based on the values of $\mathcal{R}(g_k^c)$, and use top $K$ triads as the candidate set $N_k^c$ for selection. We perform experiments with different $K$, fixing $M = 10$. Table 6 shows the results. We notice that when using smaller $K$, the selection time drops quickly but the performance of our model degrades little. This illustrates our acceleration solution is efficient and effective. Besides, the reason for performance dropping is that the total diversities of the triad candidates decreases.

## 5 CONCLUSION

In this paper, we put forward a general framework, OTGNet, to investigate open temporal graph. We devise a novel message passing mechanism based on information bottleneck to extract class-agnostic knowledge for aggregation, which can address heterophily propagation issue. To overcome catastrophic forgetting, we propose to select representative triads to memorize knowledge of old classes, and design a new value function to realize the selection. Experimental results on three real-world datasets demonstrate the effectiveness of our method.

## 6 ACKNOWLEDGEMENTS

This work was supported by the National Natural Science Foundation of China (NSFC) under Grants 62122013, U2001211.

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

# A APPENDIX

## A.1 DERIVATION OF OUR INFORMATION BOTTLENECK OBJECTIVE

Our message passing mechanism is motivated by traditional information bottleneck (Alemi et al., 2016; Tishby et al., 2000). The objective of traditional information bottleneck is $\max_Z I(Z, Y) - \beta I(Z, X)$, which attempts to maximize the mutual information between label $Y$ and latent representation $Z$, and minimize the mutual information between input feature $X$ and latent representation $Z$. Different from that, we intend to extract class-agnostic information from node embeddings. Thus, we aim to minimize the mutual information between node label $Y$ and class-agnostic representation $Z(t)$, and maximize the mutual information between input embedding $X(t)$ and class-agnostic representation $Z(t)$. Our objective could be written as: $J_{IB} = \min_{Z(t)} I(Z(t), Y) - \beta I(Z, X(t))$. First, we give a proof of the upper bound of $I(Z(t), Y)$, motivated by Cheng et al. (2020). Let $I_{club}(Z(t), Y) = \mathbb{E}_{p(Z(t), Y)}[\log p(y|z(t))] - \mathbb{E}_{p(Z(t))}\mathbb{E}_{p(Y)}[\log p(y|z(t))]$. Let $o = I_{club}(Z(t), Y) - I(Z(t), Y)$, then we have:

$$
\begin{aligned}
o &= \int dz(t) dy p(z(t), y) \log p(y|z(t)) - \int dz(t) p(z(t)) \int dy p(y) \log p(y|z(t)) \\
&\quad - \int dz(t) dy p(z(t), y) \log \frac{p(y|z(t))}{p(y)} \\
&= \int dz(t) dy p(z(t), y) \log p(y) - \int dz(t) p(z(t)) \int dy p(y) \log p(y|z(t)) \\
&= \int dy p(y) \log p(y) - dy p(y) \int dz(t) p(z(t)) \log p(y|z(t)) \\
&= \int dy p(y)(\log p(y) - \int dz(t) p(z(t)) \log p(y|z(t))).
\end{aligned}
\tag{13}
$$

Since $log(\cdot)$ is a concave function, according to Jensen's Inequality (Kian, 2014), we have:

$$
\log p(y) - \int dz p(z(t)) \log p(y|z(t)) = \log(\int dz(t) p(z(t)) p(y|z(t))) - \int dz(t) p(z(t)) \log p(y|z(t)) \geq 0.
\tag{14}
$$

Then, we have:

$$
o = \int dy p(y)(\log p(y) - \int dz(t) p(z(t)) \log p(y|z(t))) \geq 0.
\tag{15}
$$

Thus, we derive the upper bound of $I(Z(t), Y)$:

$$
I(Z(t), Y) \leq I_{club}(Z(t), Y) = \mathbb{E}_{p(Z(t), Y)}[\log p(y|z(t))] - \mathbb{E}_{p(Z(t))}\mathbb{E}_{p(Y)}[\log p(y|z(t))].
\tag{16}
$$

Since $p(y|z(t))$ is unknown, we introduce a variational approximation distribution $q_\mu(y|z(t))$ to approximate $p(y|z(t))$, following Cheng et al. (2020).

Next, we give a proof to the lower bound of $I(X(t), Z(t))$, based on Belghazi et al. (2018). According to Donsker-Varadhan representation (Donsker & Varadhan, 1975), we know:

$$
\begin{aligned}
I(X(t), Z(t)) &= KL(p(X(t), Z(t)), p(X(t))p(Z(t))) \\
&= \sup_{T:\Omega \to R} \mathbb{E}_{p(X(t), Z(t))}[T] - \log(\mathbb{E}_{p(X(t))p(Z(t))}[e^T]),
\end{aligned}
\tag{17}
$$

where $\Omega = \mathcal{X} \times \mathcal{Z}$ is the input space. Let $\mathcal{F}$ be any class of functions $T : \Omega \to R$, we have:

$$
KL(p(X(t), Z(t)), p(X(t))p(Z(t))) \geq \sup_{T \in \mathcal{F}} \mathbb{E}_{p(X(t), Z(t))}[T] - \log(\mathbb{E}_{p(X(t))p(Z(t))}[e^T]).
\tag{18}
$$

We could choose $\mathcal{F}$ to be the family of functions $T_\psi : \mathcal{X} \times \mathcal{Z} \to \mathbb{R}$ parameterized by a neural network $\psi$:

$$
KL(p(X(t), Z(t)), p(X(t))p(Z(t))) \geq \sup_\psi \mathbb{E}_{p(X(t), Z(t))}[T_\psi] - \log(\mathbb{E}_{p(X(t))p(Z(t))}[e^{T_\psi}]).
\tag{19}
$$

Thus, we have:

$$
I(X(t), Z(t)) \geq I_{mine}(X(t), Z(t)) = \sup_\psi \mathbb{E}_{p(X(t), Z(t))}[T_\psi] - \log(\mathbb{E}_{p(X(t))p(Z(t))}[e^{T_\psi}]).
\tag{20}
$$

Therefore, we could derive an upper bound of $J_{IB}$:

$$
\begin{aligned}
J_{IB} \leq \mathcal{L}_{IB} &= I_{club}(Z(t), Y) - \beta I_{mine}(X(t), Z(t)) \\
&= \mathbb{E}_{p(Z(t),Y)}[\log q_\mu(y|z(t))] - \mathbb{E}_{p(Z(t))}\mathbb{E}_{p(Y)}[\log q_\mu(y|z(t))] \\
&\quad - \beta(\sup_\psi \mathbb{E}_{p(X(t),Z(t))}[T_\psi] - \log(\mathbb{E}_{p(X(t))p(Z(t))}[e^{T_\psi}])).
\end{aligned} \tag{21}
$$

In order to minimize $J_{IB}$, we attempt to minimize its upper bound, i.e., $\mathcal{L}_{IB}$, and utilize the Monte Carlo sampling to approximate the expectations in $\mathcal{L}_{IB}$, motivated by (Wan et al., 2021; Alemi et al., 2016). Therefore, the final loss function of $\mathcal{L}_{IB}$ can be expressed as:

$$
\begin{aligned}
\mathcal{L}_{IB} = &\frac{1}{|S_d|} \sum_{i \in S_d} \left[ \log q_\mu(y_i|z_i(t)) - \frac{1}{|S_d|} \sum_{j \in S_d} \log q_\mu(y_j|z_i(t)) \right] \\
&- \frac{\beta}{|S_d|} \sum_{i \in S_d} \left[ T_\psi(x_i(t), z_i(t)) - \log \left( \frac{1}{|S_d|} \sum_{j \in S_d} e^{T_\psi(x_i(t), z_j(t))} \right) \right],
\end{aligned} \tag{22}
$$

where $S_d$ is a batch of nodes and $|S_d|$ is the size of $S_d$. $x_i(t)$, $z_i(t)$ are the embedding, the class-agnostic representation of node $i$ respectively at time-stamp $t$. $y_i$ is the label of node $i$.

## A.2 DERIVATION OF TRIAD INFLUENCE FUNCTION

In this section, we prove that our triad influence function $\mathcal{I}_{loss}(g_k^c, \theta)$ could estimate the influence of the triad $g_k^c$ on the model performance for class $k$. The triad influence function $\mathcal{I}_{loss}(g_k^c, \theta)$ is defined as:

$$
\mathcal{I}_{loss}(g_k^c, \theta) = -\nabla_\theta \mathcal{L}(G_k, \theta)^\top H_\theta^{-1} \nabla_\theta \mathcal{L}(g_k^c, \theta), \tag{23}
$$

where $\theta$ is the parameter of the model and $G_k$ is the training node set of class $k$. $H_\theta$ is the Hessian matrix. $\nabla_\theta \mathcal{L}(g_k^c, \theta)$, $\nabla_\theta \mathcal{L}(G_k, \theta)$ are the gradient of loss to $g_k^c$ and $G_k$, respectively.

The basic idea of the influence function (Cook & Weisberg, 1980; Koh & Liang, 2017) is to estimate the parameter change if a training sample is upweighted by some small $\varepsilon$ ($\varepsilon \to 0$). Thus, we add a small weight $\varepsilon$ on three nodes of the triad $g_k^c$ in the loss function $\mathcal{L}(\theta)$. The new loss function could be written as:

$$
\begin{aligned}
\mathcal{L}_{\varepsilon, g_k^c}(\theta) &= \arg\min_\theta \sum_{v \in G_k} l(v, \theta) + \varepsilon \sum_{v \in g_k^c} l(v, \theta) \\
&= \mathcal{L}(G_k, \theta) + \varepsilon \mathcal{L}(g_k^c, \theta),
\end{aligned} \tag{24}
$$

where $l(v, \theta)$ is the loss of node $v$. With the new loss function, the parameter of model is changed to $\hat{\theta}_{\varepsilon, g_k^c} = \arg\min_\theta \mathcal{L}_{\varepsilon, g_k^c}(\theta)$.

According to Cook & Weisberg (1980), we know that the influence of upweighting $\varepsilon$ could be evaluated by $\left. \dfrac{d\hat{\theta}_{\varepsilon, g_k^c}}{d\varepsilon} \right|_{\varepsilon=0}$.

Since the new loss function (24) is minimized by $\hat{\theta}_{\epsilon, g_k^c}$, we examine the first-order optimality condition:

$$
0 = \nabla_\theta \mathcal{L}(G_k, \theta) + \varepsilon \nabla_\theta \mathcal{L}(g_k^c, \theta). \tag{25}
$$

Then, since we have $\hat{\theta}_{\varepsilon, g_k^c} \to \theta$ as $\varepsilon \to 0$, we perform a Taylor expansion on the right-hand side of Eq. (25) and the higher order infinitesimal $o(\hat{\theta}_{\varepsilon, g_k^c} - \theta)$ terms are dropped :

$$
0 \approx [\nabla_\theta \mathcal{L}(G_k, \theta) + \epsilon \nabla_\theta \mathcal{L}(g_k^c, \theta)] + [\nabla_\theta^2 \mathcal{L}(G_k, \theta) + \epsilon \nabla_\theta^2 \mathcal{L}(g_k^c, \theta)](\hat{\theta}_{\varepsilon, g_k^c} - \theta). \tag{26}
$$

From Eq. (26), we could derive that

$$
\hat{\theta}_{\varepsilon, g_k^c} - \theta \approx -[\nabla_\theta \mathcal{L}(G_k, \theta) + \epsilon \nabla_\theta \mathcal{L}(g_k^c, \theta)][\nabla_\theta^2 \mathcal{L}(G_k, \theta) + \epsilon \nabla_\theta^2 \mathcal{L}(g_k^c, \theta)]^{-1}. \tag{27}
$$

Because $\theta$ minimizes $\mathcal{L}(G_k, \theta)$, $\nabla_\theta \mathcal{L}(G_k, \theta)$ equals 0. By dropping the higher order infinitesimal $o(\varepsilon)$ terms, we have:

$$
\hat{\theta}_{\varepsilon, g_k^c} - \theta \approx -[\nabla_\theta^2 \mathcal{L}(G_k, \theta)]^{-1} \nabla_\theta \mathcal{L}(g_k^c, \theta)\varepsilon. \tag{28}
$$

We denote the $H_\theta = \nabla_\theta^2 \mathcal{L}(G_k, \theta)$, and thus have:

$$\hat{\theta}_{\varepsilon, g_k^c} - \theta \approx -H_\theta^{-1} \varepsilon \nabla_\theta \mathcal{L}(g_k^c, \theta). \tag{29}$$

Then, we could estimate the change of parameters influenced by a triad as follows:

$$\left. \frac{\mathrm{d}\, \hat{\theta}_{\varepsilon, g_k^c}}{\mathrm{d}\, \varepsilon} \right|_{\varepsilon=0} = \left. \frac{\hat{\theta}_{\varepsilon, g_k^c} - \theta}{\varepsilon} \right|_{\varepsilon=0} = \left. \frac{-\varepsilon H_\theta^{-1} \nabla_\theta \mathcal{L}(g_k^c, \theta)}{\varepsilon} \right|_{\varepsilon=0} \tag{30}$$
$$= -H_\theta^{-1} \nabla_\theta \mathcal{L}(g_k^c, \theta).$$

Finally, we could derive the triad influence function $\mathcal{I}_{loss}(g_k^c, \theta)$ by the chain rule:

$$\mathcal{I}_{loss}(g_k^c, \theta) = \left. \frac{\mathrm{d}\, \mathcal{L}(G_k, \theta_{\varepsilon, g_k^c})}{\mathrm{d}\, \varepsilon} \right|_{\varepsilon=0} = \nabla_\theta \mathcal{L}(G_k, \theta)^\top \left. \frac{\mathrm{d}\, \hat{\theta}_{\varepsilon, g_k^c}}{\mathrm{d}\, \varepsilon} \right|_{\varepsilon=0} \tag{31}$$
$$= -\nabla_\theta \mathcal{L}(G_k, \theta)^\top H_\theta^{-1} \nabla_\theta \mathcal{L}(g_k^c, \theta).$$

### A.3   PROOF OF MONOTONICITY AND SUBMODULARITY

Our proposed value function is defined as:

$$F(S_k^c) = \sum_{g_{k,i}^c \in S_k^c} \nabla_\theta \mathcal{L}(G_k, \theta)^\top H_\theta^{-1} \nabla_\theta \mathcal{L}(g_{k,i}^c, \theta) + \gamma \frac{\left| \bigcup_{g_{k,i}^c \in S_k^c} \mathcal{C}_{g_{k,i}^c} \right|}{|N_k^c|}, \tag{32}$$

where $\mathcal{C}_{g_{k,i}^c} = \{ g_{k,j}^c \mid \|\bar{x}(g_{k,j}^c) - \bar{x}(g_{k,i}^c)\|_2 \leq \delta, g_{k,j}^c \in N_k^c \}$ and $N_k^c$ is the set containing all triads with positive $\mathcal{R}(g_k^c)$. As stated before, finding a fixed size set $S_k^c$ ($S_k^c \subseteq N_k^c$) that maximizes $F(S_k^c)$ is NP-hard due to the combinatorial complexity. Thus, we first prove that our value function $F$ is monotone and submodular. Then our optimization problem could be solved by a greedy algorithm with an approximation ratio guarantee according to Krause & Golovin (2014).

**Definition 1.** (Monotonicity) A function $f : 2^N \to \mathbb{R}$ is monotone if for $\forall A \subseteq B \subseteq N$, it holds that $F(A) \leq F(B)$.

**Lemma 1.** Our value function $F$ in Eq. (32) is monotone.

*Proof.* We define two triad sets $A, B$ that satisfy $A \subseteq B \subseteq N_k^c$. Let $\Delta = F(B) - F(A)$. We have:

$$\Delta = \sum_{g_{k,i}^c \in B} \nabla_\theta \mathcal{L}(G_k, \theta)^T H_\theta^{-1} \nabla_\theta \mathcal{L}(g_{k,i}^c, \theta) - \sum_{g_{k,i}^c \in A} \nabla_\theta \mathcal{L}(G_k, \theta)^T H_\theta^{-1} \nabla_\theta \mathcal{L}(g_{k,i}^c, \theta)$$

$$+ \frac{\left| \bigcup_{g_{k,i}^c \in B} \mathcal{C}_{g_{k,i}^c} \right|}{|N_c|} - \frac{\left| \bigcup_{g_{k,i}^c \in A} \mathcal{C}_{g_{k,i}^c} \right|}{|N_c|}$$

$$= \nabla_\theta \mathcal{L}(G_k, \theta)^T H_\theta^{-1} \Big( \sum_{g_{k,i}^c \in B} \nabla_\theta \mathcal{L}(g_{k,i}^c, \theta) - \sum_{g_{k,i}^c \in A} \nabla_\theta \mathcal{L}(g_{k,i}^c, \theta) \Big)$$

$$+ \frac{\left| \bigcup_{g_{k,i}^c \in B} \mathcal{C}_{g_{k,i}^c} \right|}{|N_c|} - \frac{\left| \bigcup_{g_{k,i}^c \in A} \mathcal{C}_{g_{k,i}^c} \right|}{|N_c|}$$

$$= \nabla_\theta \mathcal{L}(G_k, \theta)^T H_\theta^{-1} \sum_{g_{k,i}^c \in T} \nabla_\theta \mathcal{L}(g_{k,i}^c, \theta) + \frac{\left| \bigcup_{g_{k,i}^c \in B} \mathcal{C}_{g_{k,i}^c} \right|}{|N_c|} - \frac{\left| \bigcup_{g_{k,i}^c \in A} \mathcal{C}_{g_{k,i}^c} \right|}{|N_c|}$$

$$\geq \frac{\left| \bigcup_{g_{k,i}^c \in B} \mathcal{C}_{g_{k,i}^c} \right|}{|N_c|} - \frac{\left| \bigcup_{g_{k,i}^c \in A} \mathcal{C}_{g_{k,i}^c} \right|}{|N_c|} \geq \frac{\left| \bigcup_{g_{k,i}^c \in A} \mathcal{C}_{g_{k,i}^c} \right|}{|N_c|} - \frac{\left| \bigcup_{g_{k,i}^c \in A} \mathcal{C}_{g_{k,i}^c} \right|}{|N_c|} = 0.$$

Thus, we have:

$$\Delta = F(B) - F(A) \geq 0. \tag{33}$$
$$\Rightarrow F(A) \leq F(B). \tag{34}$$

$\square$

**Definition 2.** (Submodularity) A function $f : 2^N \rightarrow \mathbb{R}$ is submodular if for $\forall A \subseteq B \subseteq N$ and $\forall x \in N \backslash B$, it holds that $F(A \cup \{x\}) - F(A) \geq F(B \cup \{x\}) - F(B)$.

**Lemma 2.** Our value function $F$ in Eq. (32) is submodular.

*Proof.* We define two triad sets $A, B$ that satisfy $A \subseteq B \subseteq N_k^c$. Let $T = B \backslash A$. Define $\Delta = (F(A \cup \{x\}) - F(A)) - (F(B \cup \{x\}) - F(B))$. Then we have:

$$\Delta = \nabla_\theta \mathcal{L}(G_k, \theta)^T H_\theta^{-1} (\sum_{g_{k,i}^c \in A \cup \{x\}} \nabla_\theta \mathcal{L}(g_{k,i}^c, \theta) - \sum_{g_{k,i}^c \in A} \nabla_\theta \mathcal{L}(g_{k,i}^c, \theta) + \sum_{g_{k,i}^c \in B \cup \{x\}} \nabla_\theta \mathcal{L}(g_{k,i}^c, \theta) - \sum_{g_{k,i}^c \in B} \nabla_\theta \mathcal{L}(g_{k,i}^c, \theta))$$

$$+ \frac{|\bigcup_{g_{k,i}^c \in A \cup \{x\}} \mathcal{C}_{g_{k,i}^c}|}{|N_k^c|} - \frac{|\bigcup_{g_{k,i}^c \in A} \mathcal{C}_{g_{k,i}^c}|}{|N_k^c|} - (\frac{|\bigcup_{g_{k,i}^c \in B \cup \{x\}} \mathcal{C}_{g_{k,i}^c}|}{|N_k^c|} - \frac{|\bigcup_{g_{k,i}^c \in B} \mathcal{C}_{g_{k,i}^c}|}{|N_k^c|}). \quad (35)$$

$$\Delta = \nabla_\theta \mathcal{L}(G_k, \theta)^T H_\theta^{-1} (\sum_{g_{k,i}^c \in \{x\}} \nabla_\theta \mathcal{L}(g_{k,i}^c, \theta) - \sum_{g_{k,i}^c \in x} \nabla_\theta \mathcal{L}(g_{k,i}^c, \theta))$$

$$+ \frac{|\bigcup_{g_{k,i}^c \in A \cup \{x\}} \mathcal{C}_{g_{k,i}^c}|}{|N_k^c|} - \frac{|\bigcup_{g_{k,i}^c \in A} \mathcal{C}_{g_{k,i}^c}|}{|N_k^c|} - (\frac{|\bigcup_{g_{k,i}^c \in B \cup \{x\}} \mathcal{C}_{g_{k,i}^c}|}{|N_k^c|} - \frac{|\bigcup_{g_{k,i}^c \in B} \mathcal{C}_{g_{k,i}^c}|}{|N_k^c|}). \quad (36)$$

$$\Delta = \frac{|\bigcup_{g_{k,i}^c \in A \cup \{x\}} \mathcal{C}_{g_{k,i}^c}|}{|N_k^c|} - \frac{|\bigcup_{g_{k,i}^c \in A} \mathcal{C}_{g_{k,i}^c}|}{|N_k^c|} - (\frac{|\bigcup_{g_{k,i}^c \in B \cup \{x\}} \mathcal{C}_{g_{k,i}^c}|}{|N_k^c|} - \frac{|\bigcup_{g_{k,i}^c \in B} \mathcal{C}_{g_{k,i}^c}|}{|N_k^c|}). \quad (37)$$

$$\Delta = \frac{|\bigcup_{g_{k,i}^c \in A \cup \{x\}} \mathcal{C}_{g_{k,i}^c}|}{|N_k^c|} - \frac{|\bigcup_{g_{k,i}^c \in A} \mathcal{C}_{g_{k,i}^c}|}{|N_k^c|} - (\frac{|\bigcup_{g_{k,i}^c \in A \cup T \cup \{x\}} \mathcal{C}_{g_{k,i}^c}|}{|N_k^c|} - \frac{|\bigcup_{g_{k,i}^c \in A \cup T} \mathcal{C}_{g_{k,i}^c}|}{|N_k^c|}).$$

$$\Delta = \frac{1}{N_k^c}(|(\bigcup_{g_{k,i}^c \in A} \mathcal{C}_{g_{k,i}^c}) \cup (\bigcup_{g_{k,i}^c \in \{x\}} \mathcal{C}_{g_{k,i}^c})| - |(\bigcup_{g_{k,i}^c \in A} \mathcal{C}_{g_{k,i}^c})|$$

$$- |(\bigcup_{g_{k,i}^c \in A} \mathcal{C}_{g_{k,i}^c}) \cup (\bigcup_{g_{k,i}^c \in T} \mathcal{C}_{g_{k,i}^c}) \cup (\bigcup_{g_{k,i}^c \in \{x\}} \mathcal{C}_{g_{k,i}^c})| + |(\bigcup_{g_{k,i}^c \in A} \mathcal{C}_{g_{k,i}^c}) \cup (\bigcup_{g_{k,i}^c \in T} \mathcal{C}_{g_{k,i}^c})|). \quad (38)$$

For convenience, we denote $\bigcup_{g_{k,i}^c \in Q} \mathcal{C}_{g_{k,i}^c} = \mathcal{C}_Q^*$. We have:

$$\Delta = \frac{1}{|N_k^c|}(|\mathcal{C}_A^* \cup \mathcal{C}_{\{x\}}^*| - |\mathcal{C}_A^*| - |\mathcal{C}_A^* \cup \mathcal{C}_{\{T\}}^* \cup \mathcal{C}_{\{x\}}^*| + |\mathcal{C}_A^* \cup \mathcal{C}_T^*|)$$

$$= \frac{1}{|N_k^c|}(|\mathcal{C}_A^*| + |\mathcal{C}_{\{x\}}^*| - |\mathcal{C}_A^* \cap \mathcal{C}_{\{x\}}^*| - |\mathcal{C}_A^*| + |\mathcal{C}_A^* \cup \mathcal{C}_{\{T\}}^* \cup \mathcal{C}_{\{x\}}^*| + |\mathcal{C}_A^* \cup \mathcal{C}_T^*|)$$

$$= \frac{1}{|N_k^c|}(|\mathcal{C}_{\{x\}}^*| - |\mathcal{C}_A^* \cap \mathcal{C}_{\{x\}}^*| - |\mathcal{C}_A^*| - |\mathcal{C}_T^*| - |\mathcal{C}_{\{x\}}^*| + |\mathcal{C}_A^* \cap \mathcal{C}_T^*|$$

$$+ |\mathcal{C}_A^* \cap \mathcal{C}_{\{x\}}^*| + |\mathcal{C}_T^* \cap \mathcal{C}_{\{x\}}^*| - |\mathcal{C}_A^* \cap \mathcal{C}_{\{T\}}^* \cap \mathcal{C}_{\{x\}}^*| + |\mathcal{C}_A^* \cup \mathcal{C}_T^*|)$$

$$= \frac{1}{|N_k^c|}(-|\mathcal{C}_A^*| - |\mathcal{C}_T^*| + |\mathcal{C}_A^* \cap \mathcal{C}_T^*| + |\mathcal{C}_T^* \cap \mathcal{C}_{\{x\}}^*| - |\mathcal{C}_A^* \cap \mathcal{C}_{\{T\}}^* \cap \mathcal{C}_{\{x\}}^*| + |\mathcal{C}_A^* \cup \mathcal{C}_T^*|)$$

$$= \frac{1}{|N_k^c|}(-|\mathcal{C}_A^*| - |\mathcal{C}_T^*| + |\mathcal{C}_A^* \cap \mathcal{C}_T^*| + |\mathcal{C}_T^* \cap \mathcal{C}_{\{x\}}^*| - |\mathcal{C}_A^* \cap \mathcal{C}_{\{T\}}^* \cap \mathcal{C}_{\{x\}}^*|$$

$$+ |\mathcal{C}_A^*| + |\mathcal{C}_T^*| - |\mathcal{C}_A^* \cap \mathcal{C}_T^*|)$$

$$= \frac{1}{|N_k^c|}(|\mathcal{C}_T^* \cap \mathcal{C}_{\{x\}}^*| - |\mathcal{C}_A^* \cap \mathcal{C}_{\{T\}}^* \cap \mathcal{C}_{\{x\}}^*|)$$

$$\geq 0. \quad (39)$$

Then, we can derive:

$$\Delta = (F(A \cup \{x\}) - F(A)) - (F(B \cup \{x\}) - F(B)) \geq 0. \quad (40)$$

$$\Rightarrow F(A \cup \{x\}) - F(A) \geq F(B \cup \{x\}) - F(B). \quad (41)$$

$\square$

### A.4 DATASET DETAILS

**Reddit Dataset**    Reddit is a large platform of topic communities where people could write and upload their posts to share their opinions. For the Reddit dataset [1], we construct a post-to-post graph, similar to Hamilton et al. (2017). We omit the top 20 largest communities, because they are large and generic default communities, which could skew the class distribution (Hamilton et al., 2017). From the rest of communities, in each month we sample 3 largest communities that doesn't appear in previous months as new classes for each task. We take the data from July to November in 2009 and construct 6 tasks based on the selected data. In the graph, the posts are regarded as nodes and their corresponding communities are regarded as node labels. When a user comments a post at time $t$, the temporal edges at timestamp $t$ will be built, connecting this post to other posts this user has commented within a week. We initialize the feature representation of a node by averaging 300-dimensional GloVe word embeddings of all comments in this post, following Hamilton et al. (2017).

**Yelp Dataset**    Yelp is a large business review website where people could upload their reviews for commenting business, and find their interested business by others' reviews. For the Yelp dataset[2], we construct a business-to-business temporal graph, in the same way as Reddit. Specifically, we take the data from 2015 to 2019, and treat the data in each year as a task, thus forming 5 tasks in total. In each year, we sample 3 largest business categories as three classes in each task. Note that the business categories in each task have never occurred in previous tasks. We regard each business as a node and set the business's category as its node label. The temporal edge will be formed, once a user reviews the corresponding two businesses within a month. We initialize the feature representation for each node by averaging 300-dimensional GloVe word embeddings of all reviews for this business.

**Taobao Dataset**    Taobao is a large online shopping platform where items (products) could be viewed and purchased by people online. For the Taobao dataset[3](Du et al., 2019), we construct an item-to-item graph, in the same way as Reddit. The data in the Taobao dataset is a 6-days promotion season of Taobao in 2018. We set the time duration for each task as 2 days. In each two days, we take the top 30 largest item categories according to the number of items as the new classes for this task. The categories in each task have never occurred in previous tasks. We regard the items as nodes and take the categories of items as the node labels. The temporal edge will be built if a user purchases two corresponding items in the promotion season. We use the 128-dimensional embedding provided by the original dataset as the initial feature of the node.

### A.5 PSEUDO-CODE OF PROPOSED METHOD

We provide the pseudo-code of our training procedure, as shown in Algorithm 2. When learning task $\mathcal{T}_i$, we input the interactions in current task and triads of previous tasks together into our proposed message passing framework. All nodes in current interactions and previous triads are used to calculate node classification loss for training. The closed triads and open triads serve as positive and negative samples to preserve evolution patterns, respectively. After learning task $\mathcal{T}_i$, we select representative triads for the classes in $\mathcal{T}_i$ and then begin to learn task $\mathcal{T}_{i+1}$. Note that our algorithm does not guarantee that the selected triads must be connected with the nodes in the new task. This is because the selected triads play two roles in our method: on one hand, when the nodes in the triads are connected with the nodes in a new task, it will propagate knowledge among these nodes by extracting class-agnostic representations; on the other hand, triads are used to preserve the knowledge of old classes, so as to avoid catastrophic knowledge forgetting when learning new classes. Even though the selected triad does not connect with the nodes in the new task, we think it still is important for learning old classes by simultaneously considering both importance and diversity.

### A.6 IMPLEMENTATION DETAILS

We perform our experiments using GeForce RTX 3090 Ti GPU. We use the Adam optimizer for training with learning rate $\eta = 0.0001$ on the Reddit dataset, learning rate $\eta = 0.005$ on the Yelp

---

[1]https://files.pushshift.io/reddit/comments/

[2]https://www.yelp.com/dataset

[3]https://tianchi.aliyun.com/dataset/dataDetail?dataId=9716

---

**Algorithm 2** OTGNet: Open Temporal Graph Neural Networks

---

**Input:** task number $L$, last time-stamp $t$, interaction set $\mathcal{E}(t)$, node label set $\mathcal{Y}(t) = \{1, 2, ..., m(t)\}$, node initial feature $x_i(0)$, epochs $N_e$, memory budget per class $M$, trade-off parameter $\rho$;
**Output:** prediction of node classes;
  1: Initialize node embeddings;
  2: Initialize triad memory buffer $S = \emptyset$;
  3: **for** each task $\mathcal{T}_i$ from $\mathcal{T}_1$ to $\mathcal{T}_L$ **do**
  4:     **for** each epoch $e$ from 1 to $N_e$ **do**
  5:         **for** each batch $b$ in epoch $e$ **do**
  6:             let $H$ be the set containing all interactions in batch $b$;
  7:             **if** $\mathcal{T}_i \neq \mathcal{T}_1$ **then**
  8:                 let $H_{pre}$ be the set containing the interactions for all triads in $S$;
  9:                 $H = H \cup H_{pre}$;
 10:             **end if**
 11:             extract class-agnostic embeddings for all interactive nodes and their neighbors in $H$;
 12:             propagate message for all interactive nodes in $H$;
 13:             calculate node classification loss $\mathcal{L}_{ce}$ for all interactive nodes in $H$;
 14:             let $\mathcal{L} = \mathcal{L}_{ce}$;
 15:             **if** $\mathcal{T}_i \neq \mathcal{T}_1$ **then**
 16:                 calculate link prediction loss $\mathcal{L}_{link}$ for all triads in $S$;
 17:                 $\mathcal{L} = \mathcal{L}_{ce} + \rho\mathcal{L}_{link}$;
 18:             **end if**
 19:             minimize the information bottleneck loss $\mathcal{L}_{IB}$;
 20:             minimize the training loss $\mathcal{L}$;
 21:         **end for**
 22:     **end for**
 23:     **for** each class $k$ in task $\mathcal{T}_i$ **do**
 24:         select representative closed triad set $S_k^o$ for class $k$ by Algorithm 1;
 25:         select representative open triad set $S_k^o$ for class $k$ by Algorithm 1;
 26:         $S = S \cup S_k^c \cup S_k^o$;
 27:     **end for**
 28:     **for** each task $\mathcal{T}_j$ from $\mathcal{T}_1$ to $\mathcal{T}_i$ **do**
 29:         evaluate the performance of our model on the unseen test data of $\mathcal{T}_j$;
 30:     **end for**
 31: **end for**

---

datasets and learning rate $\eta = 0.001$ on the Taobao datasets. For all baselines and our method, we train each new task until convergence, and then evaluate the performance of the model on current task and all previous tasks. For the Reddit dataset and the Yelp dataset, we train each task 500 epochs. For the Taobao dataset, we train each task 100 epochs. We set the dropout rate to 0.5 on all the datasets. The node classification head is a two-layer MLP with hidden size 128. The selected triad pairs per class $M$ is set to 10 on all datasets. The sub-network extracting class-agnostic information is a two-layer MLP with hidden size 100. Note that we do not use a whole graph in the forward pass computation. For replaying the triads, we sample 5 neighbors of each node in the triad for forward propagation, motivated by the sampling strategy in TGAT and TGN. For the nodes on the current new task, we also sample 5 neighbor nodes (maybe from old class nodes) for each node to aggregate the neighborhood information.

## A.7 Additional Experiments

**Forgetting Analysis of Different Task Numbers.** We study the average forgetting (AF) changes of different methods along with the increased tasks. As stated in the main body of this paper, AF measures the decreasing extent of model performance on previous tasks compared to the best ones. Note that AF does not count the last task since the forgetting for the last task has not yet happened. As shown in Figure 5, AF of our method is generally smaller than that of other methods. This indicates that our method generally suffers from less catastrophic forgetting than other methods.

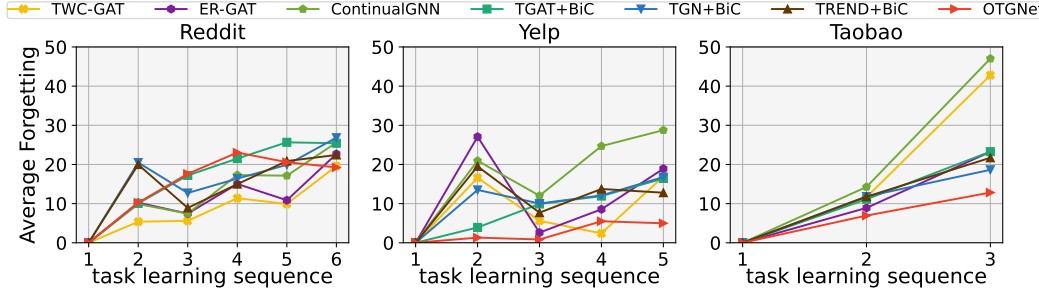

Figure 5: The changes of average forgetting (AF) (%) on three datasets with the increased tasks.

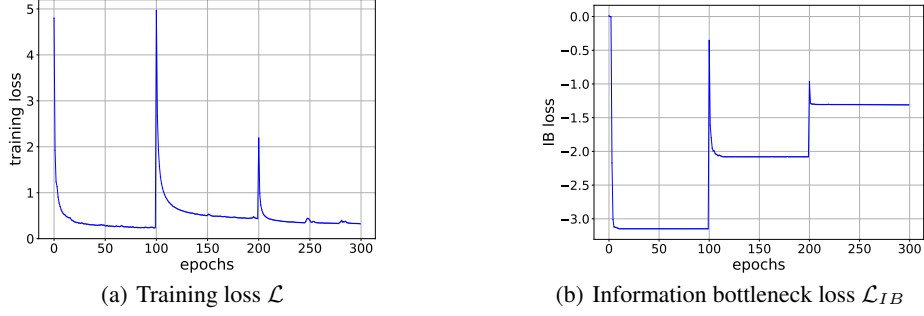

Figure 6: Convergence analysis. A new task is added for every 100 epochs.

**Convergence Analysis.** We analyze the convergence of our method. We plot the loss curves (including the total training loss $\mathcal{L}$ and the information bottleneck loss $\mathcal{L}_{IB}$ on the largest dataset, Taobao. As shown in Figure 6, our method can be eventually convergent when learning for each task.

**Sensitivity Analysis of $M$.** We analyze the influence of the number $M$ of selected triads when fixing $K = 1000$. As shown in Figure 7(a)(b), it could be observed that when we fix $K = 1000$, our method obtains good performance when $M \geq 10$. This is because with a relatively large $K$, we could select not only important but also diverse triads to preserve knowledge for achieving good performance. Thus, we set $M = 10$ throughout the experiment.

**Sensitivity Analysis of $\rho$.** We analyze the sensitiveness of $\rho$. $\rho$ is the hyper-parameter on the link prediction loss for evolution pattern preservation. As shown in Figure 8(a)(b), we observe that our method is not sensitive to $\rho$ in a relatively large range.

**Sensitivity Analysis of $\gamma$.** We analyze the sensitivity of $\gamma$ in our method. Recall that $\gamma$ is a trade-off parameter for balancing the contributions between diversity and importance when selecting representative triads. As shown in Figure 9, our method is not sensitive to $\gamma$ in a relatively large range.

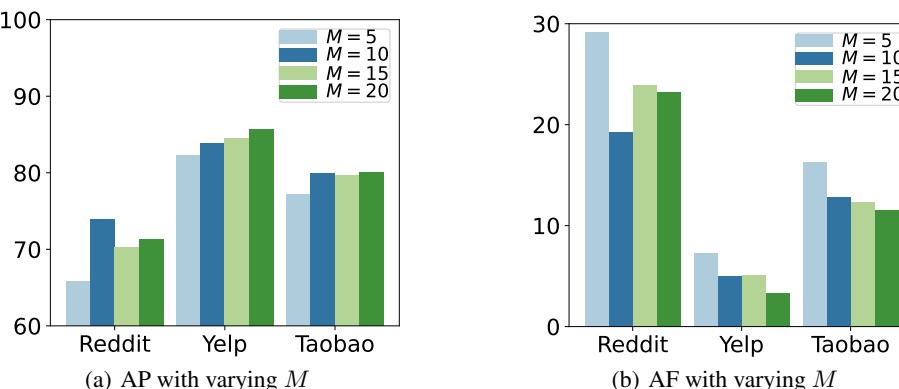

Figure 7: The sensitivity of $M$ in our method on three datasets

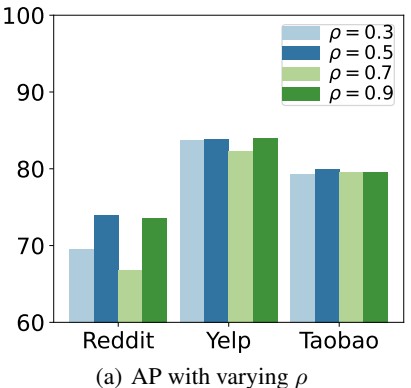
(a) AP with varying $\rho$

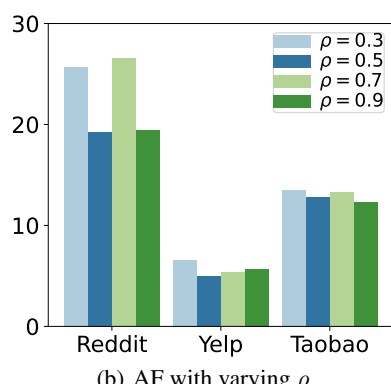
(b) AF with varying $\rho$

Figure 8: The sensitivity of $\rho$ in our method on three datasets

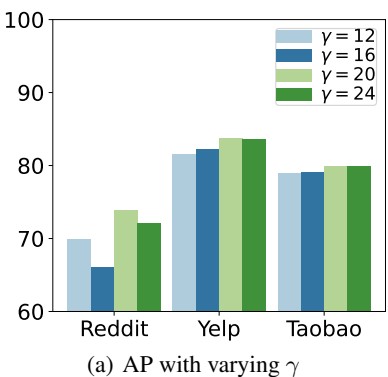
(a) AP with varying $\gamma$

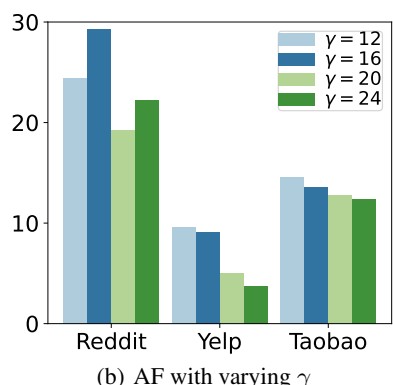
(b) AF with varying $\gamma$

Figure 9: The sensitivity of $\gamma$ in our method on three datasets.

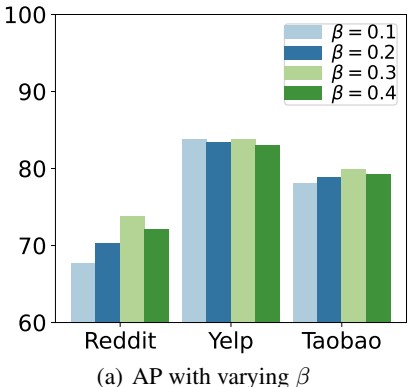
(a) AP with varying $\beta$

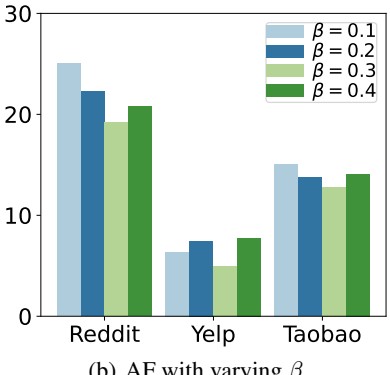
(b) AF with varying $\beta$

Figure 10: The sensitivity of $\beta$ in our method on three datasets

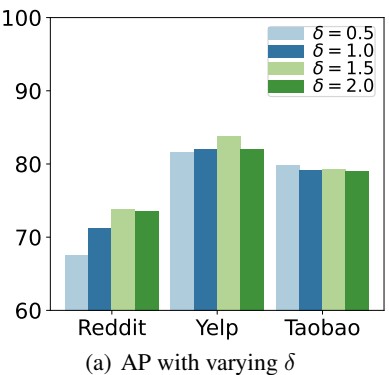 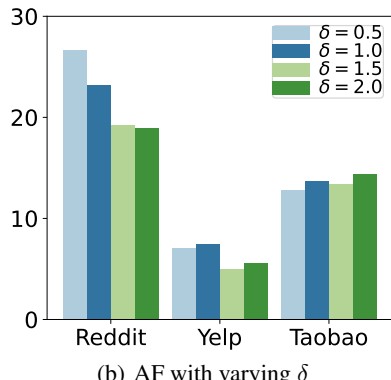

(a) AP with varying $\delta$       (b) AF with varying $\delta$

Figure 11: The sensitivity of $\delta$ in our method on three datasets

**Sensitivity Analysis of $\beta$.** We analyze the sensitivity of $\beta$ in our method. $\beta$ is the Lagrange multiplier in the objective of information bottleneck. As shown in Figure 10, we observe that our method has stable performance when changing $\beta$ in a certain range.

**Sensitivity Analysis of $\delta$.** We further analyze the sensitivity of $\delta$ in our method. $\delta$ is a hyperparameter for measuring the diversity of a triad set. As shown in Figure 11, our method is not sensitive to $\delta$ in a relatively large range.

## A.8 AP AND AF FOR EACH TASK.

Here we provide the AP and AF metric for each task of our method and two baselines (TGAT+BiC, TGN+BiC) which generally perform well among all baselines. As shown in Table 7 8 9, our method generally outperforms two baselines for most tasks.

Table 7: AP and AF for each task on the Reddit dataset.

| Method | Task 1 | | Task 2 | | Task 3 | | Task 4 | | Task 5 | | Task 6 | |
|---|---|---|---|---|---|---|---|---|---|---|---|---|
| | AP | AF | AP | AF | AP | AF | AP | AF | AP | AF | AP | AF |
| TGAT+BiC | 50.52 | 23.96 | 47.18 | 23.59 | 42.31 | 21.15 | 63.03 | 26.86 | 46.18 | 31.53 | 78.46 | - |
| TGN+BiC | 52.26 | 22.22 | 56.34 | 9.15 | 35.34 | 29.09 | 63.03 | 30.05 | 31.85 | 43.63 | 80.15 | - |
| OTGNet | 61.02 | 15.80 | 71.83 | 20.25 | 39.30 | 45.43 | 83.38 | 13.16 | 91.72 | 1.59 | 95.97 | - |

Table 8: AP and AF for each task on the Yelp dataset.

| Method | Task 1 | | Task 2 | | Task 3 | | Task 4 | | Task 5 | |
|---|---|---|---|---|---|---|---|---|---|---|
| | AP | AF | AP | AF | AP | AF | AP | AF | AP | AF |
| TGAT+BiC | 71.61 | 9.76 | 78.89 | 9.63 | 73.68 | 21.84 | 62.23 | 24.46 | 87.25 | - |
| TGN+BiC | 55.50 | 23.07 | 79.63 | 7.04 | 74.47 | 22.63 | 72.83 | 14.40 | 87.45 | - |
| OTGNet | 76.38 | 0.26 | 88.89 | 0.74 | 83.95 | 10.79 | 79.62 | 8.15 | 90.08 | - |

Table 9: AP and AF for each task on the Taobao dataset.

| Method | Task 1 | | Task 2 | | Task 3 | |
|---|---|---|---|---|---|---|
| | AP | AF | AP | AF | AP | AF |
| TGAT+BiC | 68.82 | 22.69 | 63.16 | 23.86 | 90.15 | - |
| TGN+BiC | 74.23 | 17.62 | 67.96 | 19.65 | 90.00 | - |
| OTGNet | 77.62 | 12.86 | 72.81 | 12.80 | 89.35 | - |

Table 10: Running time (hours) comparison with baselines.

| Method | Reddit | | | Yelp | | | Taobao | | |
|---|---|---|---|---|---|---|---|---|---|
| | AP | AF | Time (h) | AP | AF | Time (h) | AP | AF | Time (h) |
| TGAT+BiC | 54.61 | 25.42 | 5.05 | 74.73 | 16.42 | 3.03 | 74.05 | 23.27 | 5.82 |
| TGN+BiC | 53.16 | 26.83 | 5.81 | 73.98 | 16.79 | 3.17 | 77.40 | 18.63 | 6.23 |
| TGAT-retrain | 75.86 | 3.71 | 26.31 | 87.64 | 0.61 | 10.07 | 83.35 | 0.78 | 32.75 |
| TGN-retrain | 77.41 | 3.34 | 31.58 | 80.83 | 3.47 | 10.80 | 81.63 | 3.64 | 35.65 |
| OTGNet | 73.88 | 19.25 | 6.78 | 83.78 | 4.98 | 3.73 | 79.92 | 12.82 | 7.81 |

## A.9 RUNNING TIME ANALYSIS

When new classes occur, if we combine all data of old classes with the data of new classes for retraining, the computational complexities will be sharply increased, and be not affordable. Here, we compare the running time of our method with the retraining methods (TGAT-retrain, TGN-retrain) and two baselines (TGAT+BiC, TGN+BiC) which generally perform well among baseline methods. For the retraining methods, we use all data of old tasks for training when learning new tasks.

As shown in in Table 10, the running time of our method is comparable to the two incremental learning baselines (TGAT+BiC, TGN+BiC), while our model outperforms them on AP and AF metric with a large margin. For the retraining methods (TGAT-retrain, TGN-retrain), we can see the running time is increased by several times. In real-world applications, new classes might frequently occur. If we use all history data for training once a new class occurs, the time consuming could be unaffordable. Note that their Average Forgetting (AF) are better than our model. This is because they use all training data for learning each time, and thus can avoid forgetting.

## A.10 LIMITATIONS AND FUTURE WORKS

The quadratic time complexity of triad selection is a limitation of our method. Although only considering partial triads could be efficient, the performance of our model would be degraded to some extent. How to develop a more efficient and effective algorithm for representative triad selection is our future work. For example, we could design a hierarchical selection policy to reduce the time complexity, or develop a divide-and-conquer method for efficient triad selection. What's more, we can design better practical approximations to the theoretical optimal solution of our value function to select representative triads.

## A.11 VISUALIZATIONS

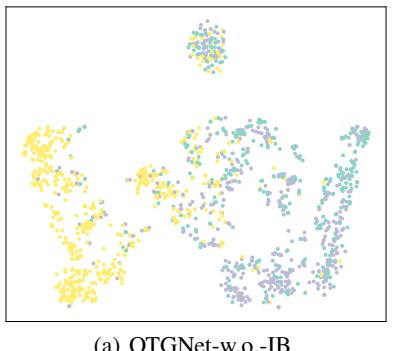
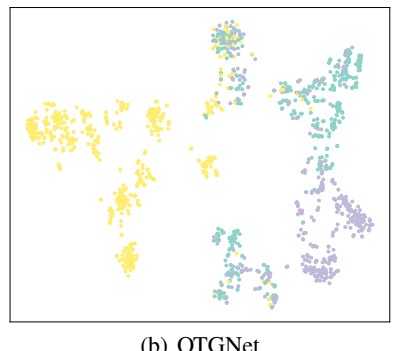

(a) OTGNet-w.o.-IB         (b) OTGNet

Figure 12: t-SNE visualization of learned node embeddings on Reddit when task 1 is finished. Different colors denote different classes.

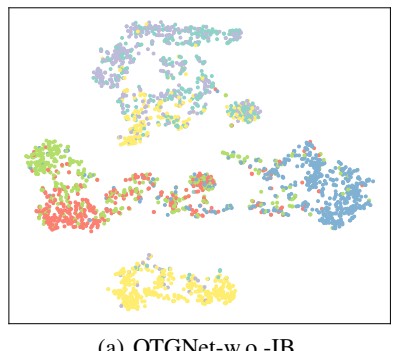 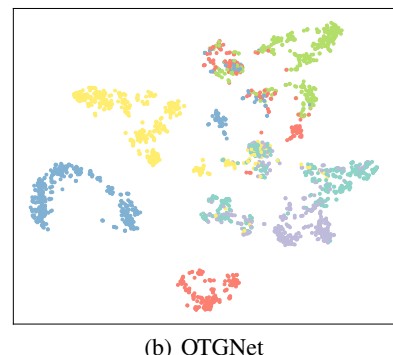

(a) OTGNet-w.o.-IB  (b) OTGNet

Figure 13: t-SNE visualization of learned node embeddings on Reddit after a new task is added. Different colors denote different classes. The new task contains 3 new classes.

To qualitatively demonstrate the effectiveness of our class-agnostic representations, we adopt t-SNE (Van der Maaten & Hinton, 2008) to visualize the learned node embeddings of our OTGNet. For comparison, we also visualize the node embeddings of OTGNet-w.o.-IB (i.e., OTGNet directly transferring the embeddings of neighbor nodes instead of class-agnostic representations). Figure 12 shows the results of their learned node embeddings on Reddit when task 1 is finished, and Figure 13 demonstrates the results after a new task is added. We can clearly observe that OTGNet possesses better representation ability by considering class-agnostic representations.

