# OpenReview forum: "Towards Open Temporal Graph Neural Networks"
_ICLR.cc/2023/Conference — ICLR 2023 notable top 5%_

### Official Review · Reviewer_FhGE · 2022-10-18

**Confidence:** 3
**Correctness:** 3
**Technical Novelty And Significance:** 3
**Empirical Novelty And Significance:** 3
**Recommendation:** 6

**Clarity, Quality, Novelty And Reproducibility:**

Clarity: The paper is largely clear with some minor ambiguity in the methods description. See weaknesses and questions.

Novelty: Overall, I think the paper presents a novel solution on an interesting problem (class-incremental GNN learning). The insights of the proposed solution are also novel. I specifically appreciate the use of class-agnostic information.

Quality: The experimental evaluations of the proposed method are extensive. OTGNet outperforms existing baselines. Ablation experiments are also extensive.

Reproducibility: This paper is not submitted with code. Also, as the methods described in this paper are rather complex and not sufficiently clearly described, I do not think that this paper is easy to reproduce.

**Strength And Weaknesses:**

Strengths:
- The problem of class-incremental GNN learning is valid and interesting. Dynamics and new classes are indeed problems in real-world graph learning.
- The observation that nodes from existing classes influence nodes from new classes due to homophily is a unique and interesting perspective. It is also shown in the experiments that this indeed poses challenges and affects the results.
- The designed methods seem novel and sound. Selecting important triads for replay to alleviate forgetting makes sense. Using influence functions to approximate the importance of triads is interesting.
- The experimental results seem very good. The improvements over existing works are significant. The ablation studies are extensive enough to demonstrate the effects of individual components.

Weaknesses and Questions:
- Some notations and words can be revised to better describe the methods. For example, $log$ should be $\log$, $exp$ should be $\exp$, $sup$ should be $\sup$, 'representative' should be 'representativeness'.
- Some details of the proposed method are not very clearly described. I list several unclear points.
    - On measuring the representativeness. We know that nodes in a graph influences the learned model in two ways, both in forward propagation (i.e. neighborhood aggregation) and backward propagation (i.e. model optimization via the loss). As far as I can see, the influence function in Eqn. 1 can only capture the latter one, i.e. whether a node appears in the loss, but not the former one, i.e. whether the node propagates its features to neighboring nodes.
    - Intuitively, the scores $\mathcal{R}(g_k^c)$ should influence each other. For example, suppose $i, j, k$ and $i, j, m$ are both close triads of class $k$, $i, j$ are both important nodes (e.g. hubs with high degrees) while $j, m$ are not. It is likely that both triads will get high scores. However, it is often sufficient to select only one of them (as only $i, j$ are important), i.e. with the presence of $i, j, k$, the score of $i, j, m$ will decrease. Is my understanding correct? If yes, how can OTGNet address for the case?
    - In the triad structure replay, it is said that 'we will replay these triads from old classes when learning new classes'. However, it is not clear how will nodes that are not in selected triads be dealt with. Specifically, will they still participate in the forward propagation? If they are dropped, there will be missing auxiliary information (e.g. nodes in the selected triads will have missing neighbors). If they are not, the training cost is still high (since a whole graph participates in forward computation). Please clarify that.
    - In Eqn. 10, 11, 12, it seems that $x_i(t), z_i(t)$ are not related to the layer. Does it imply that OTGNet only supports one layer of TGAT?
- I appreciate the extensiveness of experiments with the following minor questions.
    - I suggest some visualization or case study on the class agnostic embeddings $z$ and $x$ to better illustrate the necessity of learning $Z$.
    - In Table 6, changing $K$ from 1000 to 100 leads to a significant performance drop. However, in Figure 7, it shows that the performance stabilizes with only $M=20$, which means that only 20 triads per class is enough for OTGNet to work. This seems counter-intuitive. From my understanding, it indicates that the triads with high $\mathcal{R}$ (top $K$) do not necessarily lead to a good selection. Is that correct?

**Summary Of The Paper:**

This paper proposes OTGNet to tackle the problem of continual learning/class-incremental learning on graphs. OTGNet consists of two components. The first component is an open/close triad selection and replay method to avoid catastrophic forgetting of existing knowledge. The second component is an information bottleneck module to decouple class-related knowledge from class-agnostic knowledge, and use the class-agnostic knowledge for propagation between nodes from new classes and existing classes. It is designed to avoid over-smoothing between existing classes and new classes. The authors perform experiments on real-world class-incremental settings. The proposed OTGNet outperforms various baselines of continual learning on graphs.

**Summary Of The Review:**

Overall, I think this paper proposes a novel and insightful solution to an important problem, with solid experimental evaluations. I recommend accepting the paper.

---

> ### Author Response · Authors · 2022-11-16
> **Response To Reviewer FhGE (Part 2)**
>
> > Q6: I suggest some visualization or case study on the class agnostic embeddings $z$  and $x$ to better illustrate the necessity of learning  $Z$.
>
> A6: Thanks for your suggestion. We have added visualization of our method in Appendix A.11. To qualitatively demonstrate the effectiveness of our class-agnostic representations, we adopt t-SNE  to visualize the learned node embeddings of our OTGNet. For comparison, we also visualize the node embeddings of OTGNet-w.o.-IB (i.e., OTGNet directly transferring the embeddings of neighbor nodes instead of class-agnostic representations). The visualization results are provided in Appendix A.11.  We can clearly observe that OTGNet possesses better representation ability by considering class-agnostic representations.
>
>
>
> > Q7: In Table 6, changing $K$ from 1000 to 100 leads to a significant performance drop. However, in Figure 7, it shows that the performance stabilizes with only $M=20$ , which means that only 20 triads per class is enough for OTGNet to work. This seems counter-intuitive. From my understanding, it indicates that the triads with high  $\mathcal{R}$ (top $K$ ) do not necessarily lead to a good selection. Is that correct?
>
> A7: Sorry for confusing you. Recall that we intend to select $M$ triads from top $K$ triad candidates with high $\mathcal{R}$ for each class.
> Table 6 shows the  performance of our acceleration solution when decreasing $K$. We could find that when fixing $M=10$,  the performance of our method drops as $K$ decreases from 1000 to 100. This is because the total diversities of the triad candidates decreases. Figure 7 shows the performance of different $M$ for each class. It could be observed that when we fix $K=1000$, our method obtains good performance when $M\geq10$. This is because with a relatively large $K$, we could select not only important but also diverse triads to preserve knowledge for achieving good performance. We have added more descriptions in the revised paper.
>
>
>
> > Q8: This paper is not submitted with code. Also, as the methods described in this paper are rather complex and not sufficiently clearly described, I do not think that this paper is easy to reproduce.
>
> A8: Thanks for your comment. We will release the code, once this paper is accepted.

---

> > ### Comment · Reviewer_FhGE · 2022-11-18
> > **Author response received.**
> >
> > Dear authors,
> >
> > I have read your response and find it informative. The response clarify my misunderstandings.
> >
> > I will take some time to read other reviews and see whether I have follow up questions ASAP.

---

> ### Author Response · Authors · 2022-11-16
> **Response To Reviewer FhGE (Part 1)**
>
> > Q1: Some notations and words can be revised to better describe the methods. For example, $log$ should be $\log$,  $exp$ should be $\exp$,  $sup$ should be $\sup$, 'representative' should be 'representativeness'.
>
> A1: Thanks for your advice. We have corrected them in the revised paper.
>
>
>
> > Q2:  On measuring the representativeness. We know that nodes in a graph influences the learned model in two ways, both in forward propagation (i.e. neighborhood aggregation) and backward propagation (i.e. model optimization via the loss). As far as I can see, the influence function in Eqn. 1 can only capture the latter one, i.e. whether a node appears in the loss, but not the former one, i.e. whether the node propagates its features to neighboring nodes.
>
> A2:  Thanks for your suggestion. Eq. (1) in our paper can estimate the change of the loss when without using a triad for training. For calculating the loss, we first aggregate neighbor information to obtain the node embedding for prediction. Therefore, Eq. (1) can implicitly consider the neighborhood information to some extent.  In our future work, we would like to explicitly model the influence of a center node to its neighbors.
>
>
>
> > Q3:  Intuitively, the scores  $\mathcal{R}(g^c_k)$ should influence each other. For example, suppose $i,j,k$ and $i,j,m$ are both close triads of class $k$,  $i,j$ are both important nodes (e.g. hubs with high degrees) while  are not. It is likely that both triads will get high scores. However, it is often sufficient to select only one of them (as only $i,j$ are important), i.e. with the presence of $i,j,k$, the score of $i,j,m$ will decrease. Is my understanding correct? If yes, how can OTGNet address for the case?
>
> A3: Yes, that is correct. Our method attempts to select triads by considering not only importance but also diversity, as expressed in Eq. (6) in our paper. Thus, when two triads having high important scores are very similar, our method can only select one of them by considering their diversities.  Table 4 shows the effectiveness of such a strategy. In Table 4,  the performance of our method will  decrease, if without considering the diversity of the triads.
>
>
>
> > Q4: In the triad structure replay, it is said that 'we will replay these triads from old classes when learning new classes'. However, it is not clear how will nodes that are not in selected triads be dealt with. Specifically, will they still participate in the forward propagation? If they are dropped, there will be missing auxiliary information (e.g. nodes in the selected triads will have missing neighbors). If they are not, the training cost is still high (since a whole graph participates in forward computation). Please clarify that.
>
> A4: Sorry for confusing you.  We do not use a whole graph in forward computation. For replaying the triads, we sample 5 neighbors of each node in the triad for forward propagation, motivated by the sampling strategy in TGAT and TGN. For the nodes on the current new task, we also sample 5 neighbor nodes (maybe from old class nodes) for each node to aggregate neighbor information. We have added more description in Appendix A.6.
>
>
>
> > Q5: In Eqn. 10, 11, 12, it seems that  $x_i(t), z_i(t)$ are not related to the layer. Does it imply that OTGNet only supports one layer of TGAT?
>
> A5: Our method supports multiple layers of network. We do not use the symbol of the layer only for writing conveniently. We have added the explanation in Page 6 of the revised paper.

---

### Official Review · Reviewer_Riiz · 2022-10-23

**Confidence:** 3
**Correctness:** 3
**Technical Novelty And Significance:** 2
**Empirical Novelty And Significance:** 3
**Recommendation:** 6

**Clarity, Quality, Novelty And Reproducibility:**

Overall, I think it's a good paper. The paper proposed a new setting for temporal graphs and proposed a new algorithm based on the task. The task is interesting and novel in the temporal graph domain. The ablation study also shows that the model can work as expectation. The paper also provides some experiments to show that their model can achieve state-of-the-art performance. The paper is technically sound. The paper needs some background to have a better understanding.

**Strength And Weaknesses:**

Strength:

This paper proposed a framework to handle the class-incremental task in the temporal graph. The Triad Structure Selection is interesting. By using the influence function, it can select the important triangles. The task is novel and interesting. To accelerate the algorithm, the authors also apply a greedy result by showing that the value function F is monotone and submodular. This is intuitive. The experiments also shows that the model compared with TGN and TGAT have a better result, the ablation result shows that all the module works well. The experiments in the appendix show that the model can learn the class-agnostic knowledge and is able to mitigate the catastrophic forgetting problems. The ablation study is solid.

Weaknesses: I have the following questions:

First, the paper mentioned the heterophily setting in the temporal graph, which can be one main issue. However, in previous research, people found that the heterophily indicates that usually, GNN(such as vanilla GCN) can't work, but the traditional pagerank-based algorithms(APPNP) or other models which have a more powerful aggregator method(such as GCNII, GGCN, or other attention-based models, which implicitly handle the heterophily by a learning-based aggregator) perform well in these heterophily tasks. I think this kind of model should also be a baseline (although I understand the setting is a temporal graph and new coming tasks, so maybe the authors can first ignore the temporal information. ). Moreover, the baselines are not enough, both TGN and TGAT are baselines 2 years ago. Plenty of temporal network models are proposed and should be compared.

Secondly, the time complexity of choosing triads is unscalable. If the testing set is large, or the graph itself is large, the algorithm can't work in a limited time. However, for temporal graphs, we usually have more edges due to the time domain. Some possible work like Causal Anonymous Walk may somehow solve this issue by random walk. And these methods also have the potential to represent more complex motives instead of triangles.

[1] Two Sides of the Same Coin: Heterophily and Oversmoothing in Graph Convolutional Neural Networks
[2] Inductive Representation Learning in Temporal Networks via Causal Anonymous Walks

**Summary Of The Paper:**

This paper proposed a new task in temporal networks and with new methods. The proposed problem is related to the new class appearing at different times.

The authors claimed there are two challenges for the current temporal network: 1) How to dynamically propagate appropriate information in an open temporal graph, where new class nodes are often linked to old class nodes. This indicates that temporal graphs are usually homophily. 2) How to avoid catastrophic knowledge forgetting over old classes when learning new classes occurred in temporal graphs. No previous temporal network model includes a class-incremental learning setting.

To handle these two challenges, the authors proposed an information bottleneck-based message-passing framework, assuming that the information of a node can be disentangled into a class-relevant and class-agnostic one.

**Summary Of The Review:**

Overall, I think this paper is borderline paper. This paper proposed a novel task. It focused more on the class-incremental learning problem. The tasks and proposed methods all focused on this topic. The temporal network setting is more like a specific task. The paper also proposed some new techniques to solve this problem but these techniques are mostly from papers from other domains. The performance is not solid enough, with only 2 baselines that are only focused on the temporal network (can't solve either class incremental problems or heterophilic graph problems).

---

> ### Author Response · Authors · 2022-11-16
> **Response To Reviewer Riiz (Part 2)**
>
> >Q3: Secondly, the time complexity of choosing triads is unscalable. If the testing set is large, or the graph itself is large, the algorithm can't work in a limited time. However, for temporal graphs, we usually have more edges due to the time domain. Some possible work like Causal Anonymous Walk may somehow solve this issue by random walk. And these methods also have the potential to represent more complex motives instead of triangles.
>
> A3: Thanks for your suggestion. On a large graph, the time consumption of choosing triads could become large. To reduce the cost, we propose an acceleration solution in Sect. 3.3 in our paper, which only selects triads from the top-K ranked triads with high $\mathcal{R}(g_k^c)$.  Table 6 lists the acceleration results. When using less candidates (e.g., $K=200$) for selection, the time drops sharply but the performance of our model degrades not too much.
>
> To further reduce the time complexity of ranking triads which is of order at most $O(d_k|\mathcal{E}_k|)$, where $|\mathcal{E}_k|$ is the number of edges between two nodes of class $k$ and $d_k$ is the max degree of nodes of class $k$, we can sample a fixed number of triads  from the graph for selection. What's more, we could also design a hierarchical selection policy or adopt some distributed  methods for acceleration.
>
> The triad structure is a fundamental element of temporal graph and its triad closure process can well capture the evolution patterns as pointed out by previous works [3, 4]. Motivated by this, we choose the triad structure as motif to preserve knowledge on an open temporal graph in this paper. It is an interesting idea to explore Causal Anonymous Walk [5] to capture more effective motives in our scenario. We will study it in our future work.
>
>
>
> References:
>
> [1] GBK-GNN: Gated bi-Kernel graph neural networks for modeling both homophily and heterophily (WWW 2022)
>
> [2] TREND: TempoRal event and node dynamics for graph representation learning (WWW 2022)
>
> [3] Mining triadic closure patterns in social networks (WWW 2014)
>
> [4] Link and triadic closure delay: Temporal metrics for social network dynamics (AAAI 2014)
>
> [5] Inductive representation learning in temporal networks via causal anonymous walks (ICLR 2021)

---

> ### Author Response · Authors · 2022-11-16
> **Response To Reviewer Riiz (Part 1)**
>
> > Q1: In previous research, people found that the heterophily indicates that usually, GNN(such as vanilla GCN) can't work, but the traditional pagerank-based algorithms(APPNP) or other models which have a more powerful aggregator method(such as GCNII, GGCN, or other attention-based models, which implicitly handle the heterophily by a learning-based aggregator) perform well in these heterophily tasks.  I think this kind of model should also be a baseline (although I understand the setting is a temporal graph and new coming tasks, so maybe the authors can first ignore the temporal information. )
>
> A1: Thanks for your suggestion. There are indeed some works which can handle the heterophily issue in static graph. Different from them, we investigate how to solve this issue in an open temporal graph. In order to further evaluate our method, we compare it with  GBK-GNN [1] proposed recently.   GBK-GNN originally handles the heterophily in static graph by two kernel feature transformation matrices to capture homophily and heterophily information between node pairs respectively.
> For a fair comparison, we modify GBK-GNN to an open temporal graph: Specifically, we create two temporal message propagation modules with separated parameters in OTGNet-w.o.-IB (OTGNet-w.o.-IB means our method directly transfers the embeddings of neighbor nodes instead of class-agnostic representations). We regard the two temporal message propagation modules as the two kernel feature transformation matrices in GBK-GNN.  When aggregating neighbor information, we sum up the message derived by these two modules with a dynamic weight, as in traditional GBK-GNN. We denote this baseline as OTGNet-GBK. and add its results into our ablation study (Table 3). The results are as follows:
>
> |            | Reddit |       | Yelp  |      | Taobao |      |
> | ---------- | ------ | ----- | ----- | ---- | ------ | ---- |
> |            | AP     | AF    | AP    | AF   | AP     | AF   |
> | OTGNet-GBK | 76.53  | 18.43 | 82.56 | 6.32 | 86.65  | 9.78 |
> | OTGNet     | 81.87  | 15.54 | 86.31 | 4.77 | 88.06  | 9.26 |
>
> From the above table, we can find that OTGNet has better performance than OTGNet-GBK. This illustrates extracting class-agnostic representations is more effective to handle the heterophily issue in an open temporal graph, compared to the baseline.
>
>
>
>  >Q2: Moreover, the baselines are not enough, both TGN and TGAT are baselines 2 years ago. Plenty of temporal network models are proposed and should be compared.
>
> A2: Thanks for your suggestion. Since TGN and TGAT are two representative models in temporal graph, we compare them with our method.  Moreover, we actually also compare  with three typical incremental learning methods which are based on static GNNs, i.e., ER-GAT, TWC-GAT, and ContinualGNN. To further verify the effectiveness of our method, we add a comparison with a temporal model TREND [2] proposed recently. The experimental results are listed in the following table:
>
> |             | Reddit |       | Yelp  |       | Taobao |       |
> | ----------- | ------ | ----- | ----- | ----- | ------ | ----- |
> |             | AP     | AF    | AP    | AF    | AP     | AF    |
> | TREND       | 55.76  | 35.82 | 65.78 | 25.70 | 62.54  | 27.78 |
> | TREND+EWC   | 60.58  | 26.30 | 72.76 | 15.31 | 70.32  | 17.87 |
> | TREND+iCaRL | 63.12  | 28.34 | 76.68 | 12.35 | 77.13  | 13.59 |
> | TREND+BiC   | 65.46  | 27.18 | 77.90 | 11.84 | 78.91  | 12.20 |
> | OTGNet      | 81.87  | 15.54 | 86.31 | 4.77  | 88.06  | 9.26  |
>
> As shown in the above table, our OTGNet still outperforms TREND and the combination of TREND and the class-incremental learning methods. We have added this result in Table 1 of the revised paper.

---

### Official Review · Reviewer_KnTc · 2022-10-24

**Confidence:** 2
**Correctness:** 2
**Technical Novelty And Significance:** 4
**Empirical Novelty And Significance:** 4
**Recommendation:** 6

**Clarity, Quality, Novelty And Reproducibility:**

Clarity need to be improved.
Quality and novelty are fine.

**Strength And Weaknesses:**

## Strength

- This paper focues on continuous open temporal graph which is lack of formal study in previous works.
- This paper proposes OTGNet which is empirically far better than similar works.

## Weakness

Some statements need to be improved for better understanding.

- The definition of $I_{loss}$ need to be improved. For example, upweight by $\epsilon$ need to be formally expanded. I understand that $\epsilon$ is the extra weighted classification loss only for traidic nodes only after reading the appendix.
- Will it be possible that Algorithm 1 select triadics that are not connected with any node in current batch or task? I think those triadics are meaningless. If Algorithm 1 can avoid this, how it is guaranteed? I do not see any step in Algorithm 1 to avoid this.
- Class-agnostic representation $Z(t)$ need to be clarified. How to really compute $L_{IB}$? I think it is infeasible to compute any expection value in the formula.
- Is neural network method the only way to get $Z(t)$? I think this optimization is quite classic, and may have non-parametric approximation. Can you confirm that there is no other approximation?

Besides, I have a minor concern of Algorithm 2:

- For every task, you will select new important triadics $S_k$ for each class and update them in the memory. Thus the triadic buckect $S$ will indeed grow without restriction. Is there any way that you can limit the maximum size of $S$? If so, how will this restriction hurts the performance?

**Summary Of The Paper:**

This paper focusing on continuous open temporal graph where new nodes and new target node labels will join with time going on. This field is lack of study, and this paper proposes OTGNet which is far better than similar works.

**Summary Of The Review:**

This paper focues on continuous open temporal graph task which is lack of study, and make a concrete contribution to the field with OTGNet which is empirically far better than similar baselines.

---

> ### Author Response · Authors · 2022-11-16
> **Response To Reviewer KnTc (Part 2)**
>
> >Q4: Is neural network method the only way to get $Z(t)$? I think this optimization is quite classic, and may have non-parametric approximation. Can you confirm that there is no other approximation?
>
> A4: Thanks for your comment. To derive $Z(t)$, we attempt to minimize the mutual information between the class-agnostic representation $Z(t)$ and node label $Y$, while maximize the mutual information between $Z(t)$ and input embedding $X(t)$. As we know, it is difficult to compute mutual information.  The exact computation of mutual information is only tractable for a discrete variable or for the variable with a known probability distribution [3].  For a continuous variable with an unknown probability distribution, there are indeed some non-parametric approximation methods for calculating the mutual information, such as maximum likelihood-ratio estimators [4] and non-parametric kernel-density estimators [5].  However, these non-parametric methods can work well for a low-dimension variable, but often have bad performance when the dimension of the variable is high [3, 6]. Recently, there are some works  exploiting the neural network to approximate the mutual information and achieving promising performance [1, 3]. Inspired by this, we also adopt the neural network for optimization.
>
>
> > Q5: minor concern: For every task, you will select new important triadics $S_k$ for each class and update them in the memory. Thus the triadic bucket  $S$ will indeed grow without restriction. Is there any way that you can limit the maximum size of $S$? If so, how will this restriction hurts the performance?
>
> A5: Thanks for your comment. In the field of computer vision, many memory-based incremental learning methods [7, 8]  select a fixed number of representative samples for each class, where  the size of memory bucket is allowed to grow as the number of new classes increases. Moreover, there are some works which fix the size of memory bucket [9, 10]. For example, when the bucket is full, the number of samples for each class can be decreased by uniformly discarding samples from each old class [9]. This could result in increasing knowledge forgetting of old classes. Different from [9], the work in [10] attempts to  dynamically assign different numbers of exemplars for each class according to the recognition difficulty of each class.  In principle, these strategies could be applied to our model. It is interesting to study them in our future work.
>
> References:
>
> [1] Multi-view information-bottleneck representation learning (AAAI 2021)
>
> [2] Deep Variational Information Bottleneck (ICLR 2017)
>
> [3] Mutual information neural estimation (ICML 2018)
>
> [4] Approximating mutual information by maximum likelihood density ratio estimation (New challenges for feature selection in data mining and knowledge discovery, PMLR, 2008)
>
> [5] Estimation of mutual information using kernel density estimators (Physical Review E 1995)
>
> [6] Efficient Estimation of Mutual Information for Strongly Dependent Variables (AISTATS 2015)
>
> [7] Topology preserving class-incremental learning (ECCV 2020)
>
> [8] Distilling causal effect of data in class-incremental learning (CVPR 2021)
>
> [9] Mnemonics training: Multi-class incremental learning without forgetting (CVPR 2020)
>
> [10] RMM: Reinforced memory management for class-incremental learning (NeurIPS 2021)

---

> ### Author Response · Authors · 2022-11-16
> **Response To Reviewer KnTc (Part 1)**
>
>  >Q1:  I understand that $\varepsilon$ is the extra weighted classification loss only for traidic nodes only after reading the appendix.
>
> A1: Thank you for pointing out this. We have added the explanation of $\varepsilon$ in Page 4 of the revised paper: $\varepsilon$ is a small weight added on the three nodes of the triad $g^c_k$ in the loss function $\mathcal{L}$.
>
>
>
>  >Q2: Will it be possible that Algorithm 1 select triadics that are not connected with any node in current batch or task? I think those triadics are meaningless. If Algorithm 1 can avoid this, how it is guaranteed? I do not see any step in Algorithm 1 to avoid this.
>
> A2: Thanks for your comment. The selected triads play two roles in our method: on one hand, when the nodes in the triads are connected with the nodes in a new task, it will propagate knowledge among these nodes by extracting class-agnostic representations; on the other hand, triads are used to preserve the knowledge of old classes, so as to avoid  catastrophic knowledge forgetting when learning new classes. Even though the selected triad does not connect with the nodes in the new task, we think it still is important for learning old classes by simultaneously considering both importance and diversity.
> Thus, our algorithm does not guarantee that the selected triads must be  connected with the nodes in the new task. We have added more details in Appendix.5 of the revised paper.
>
>
>
> >Q3: Class-agnostic representation $Z(t)$ need to be clarified. How to really compute $\mathcal{L}_{IB}$? I think it is infeasible to compute any expection value in the formula.
>
> A3: Thanks for your comment.  $Z(t)$ is a random variable of class-agnostic representations at time-stamp $t$. $z(t)$ is the class-agnostic representation of  a node at time-stamp $t$, and $z(t)$ is an instance of $Z(t)$.
>
> In order to minimize the objective function $J_{IB}$, we attempt to minimize its upper bound, i.e., $L_{IB}$,  and utilize the Monte Carlo sampling to approximate the expectations in $L_{IB}$ , motivated by  [2, 3]. Therefore, the final loss function $L_{IB}$ can be expressed as:
>
> $$
> L_{IB} = \frac{1}{|S_{d}|}\sum_{i\in S_d}\ [\ \log q_\mu(y_i|z_i(t))-\frac{1}{|S_{d}|}\sum_{j\in S_d}{\log q_\mu(y_j|z_i(t))}\ ]\ \\
> -\frac{\beta}{|S_{d}|}\sum_{i\in S_d}\ [\ T_\psi(x_i(t),z_i(t))-\log{(\frac{1}{|S_{d}|}\sum_{j\in S_d}e^{T_\psi(x_i(t),z_j(t))})}\ ],
> $$
>
>
> where $S_d$ is a batch of nodes and $|S_d|$ is the size of $S_d$. $x_i(t)$ and $z_i(t)$ are the embedding and the class-agnostic representation of node $i$  at time-stamp $t$, respectively. $y_i$ is the label of node $i$. We have added more details in Appendix A.1.

---

### Official Review · Reviewer_WeDg · 2022-10-24

**Confidence:** 4
**Correctness:** 4
**Technical Novelty And Significance:** 4
**Empirical Novelty And Significance:** 4
**Recommendation:** 8

**Clarity, Quality, Novelty And Reproducibility:**

This paper is well structured and in general easy to follow. The evaluation is thorough and extensive but the experimentation setting needs further elaborations. The proposed solution is novel and original.

**Strength And Weaknesses:**

Strength:
1. The problem of learning open temporal graph is of great practical value as in many applications where graphs are updated consistently with both new and old node types. The algorithm proposed in the paper clearly improves the state of art temporal GNN algorithms under this setting.
2. The proposed methods are well motivated, and are supported with both theoretical and empirical results.
3. The proposed algorithm is very generalizable. It works with basically any temporal GNN model with no or little modifications.

Weakness:
1. The experimentation process needs further elaboration, especially on how the baseline models are trained and tuned. For example, there might be trade-off between AP and AF depending on how many epochs the model trains on the new task.
2. While the loss function encourage class-agnostic representation, to what extend the learned representation is class-agnostic can be better depicted besides the ablation study measured by model performances. For example, suppose we cluster the embeddings, will the clusters correlate with classes?

**Summary Of The Paper:**

The paper focuses on learning GNNs on open temporal graph where new nodes with novel classes are also added to the graph. The proposed method solves two major issues that existing temporal GNN methods have under this setting: learning with added heterophily and catastrophic forgetting. To solve the catastrophic forgetting, a data selection process motivated by the fluence function is considered which is defined to maximize the representativeness of the selections. The author proposes an information bottleneck based loss function to encourage the model to learn the class agnostic representations. Experimentations on real-world benchmark suggests this algorithm works with state of art temporal GNN and performs favorably.

**Summary Of The Review:**

This paper provides a neat and novel solution to the open temporal graph, a setting is of great practical value but not actively researched. The algorithm's correctness is proved both theoretically and empirically.

---

> ### Author Response · Authors · 2022-11-16
> **Response To Reviewer WeDg**
>
>  >Q1: The experimentation process needs further elaboration, especially on how the baseline models are trained and tuned. For example, there might be trade-off between AP and AF depending on how many epochs the model trains on the new task.
>
> A1: Thanks for your suggestion. We have added more details in Appendix A.6: For all baselines and our method, we train each new task until convergence, and then test the performance of the model  on current task and all previous tasks.  For the Reddit dataset, we train each task 500 epochs. For the Yelp and Taobao datasets, we train each task 100 epochs because we find that 100 epochs is enough for the convergence on Yelp and Taobao.
>
>  >Q2:  While the loss function encourage class-agnostic representation, to what extend the learned representation is class-agnostic can be better depicted besides the ablation study measured by model performances. For example, suppose we cluster the embeddings,  will the clusters correlate with classes?
>
> A2:  Thanks for your suggestion. We have added visualization of our method in Appendix A.11. To qualitatively demonstrate the effectiveness of our class-agnostic representations, we adopt t-SNE  to visualize the learned node embeddings of our OTGNet. For comparison, we also visualize the node embeddings of OTGNet-w.o.-IB (i.e., OTGNet directly transferring the embeddings of neighbor nodes instead of class-agnostic representations). The visualization results are provided in Appendix A.11. We can clearly observe that OTGNet possesses better representation ability by considering class-agnostic representations.

---

> > ### Comment · Reviewer_WeDg · 2022-11-18
> > **Thanks for the update**
> >
> > I appreciate the efforts from the authors to address the issues. The responses are satisfactory.

---

### Author Response · Authors · 2022-11-16
**General Response**

We thank all the reviewers for their insightful and constructive feedback. We really appreciate all the four reviewers thought our method to be "novel" and  "having good empirical results".

We have made point-to-point response to the comments of each reviewer  and uploaded our revised version.  Finally, we once again thank all reviewers for their insightful comments which are very helpful for improving the quality of our paper.

---

### Author Response · Authors · 2023-02-24
**Camera-ready Version**

We have uploaded the camera-ready version now. While preparing the camera-ready, we found a bug in the evaluation of each task’s performance. We have corrected this bug and uploaded the new experimental results in the camera-ready version. After fixing the bug, our method still outperforms baseline methods in a large margin (e.g. around 20% AP gain on Reddit). Based on the new results, the conclusion in this paper is kept unchanged. Besides, we added some descriptions to make our method more clarified. Our codes are now available at: https://github.com/tulerfeng/OTGNet .

---

### Decision · Program_Chairs · 2023-01-20

**Decision:**

Accept: notable-top-5%

**Justification For Why Not Higher Score:**

N/A

**Justification For Why Not Lower Score:**

I think that the work presented here is not only very nicely executed, with strong empirical results, but also that the setup considered by the authors has potential to be game-changing in the future, for both industrial and scientific applications. It is my view that it deserves a significant amplification at ICLR in terms of 'screen-time', and hence recommend it for a long oral.

**Metareview: Summary, Strengths And Weaknesses:**

The authors pose a novel and important setting for dynamic graph network representation learning: handling an open set of classes over time. This is likely a highly important setting for real-world industrial utility of these methods---real-world networks (such as social networks) are not only highly dynamic but highly variable, and responses to novel kinds of events may well necessitate novel classes of users. The proposed approach, based on disentangling a node's representation into class-agnostic and class-specific knowledge, is certainly sensible. But most importantly, the method empirically performs very well compared to basically any other method currently available. After the rebuttal, all reviewers concur that this is very solid work that should be accepted at ICLR. The main remaining issue appears to be the clarity of the method description, as some of the reviews have pointed out that the method is not clearly described (concerning aspects such as, e.g., selecting triads, influence function, etc.).

**Note From Pc:**

if the above contains the word "oral" or "spotlight" please see: "oral" presentation means -> notable-top-5% and "spotlight" means -> notable-top-25%. As stated in our emails, we are disassociating presentation type from AC recommendations